# Online Weighted Paging with Unknown Weights

**Orin Levy**[*]
Tel-Aviv University
orinlevy@mail.tau.ac.il

**Noam Touitou**
Amazon Science
noamtwx@gmail.com

**Aviv Rosenberg**[†]
Google Research
avivros007@gmail.com

## Abstract

Online paging is a fundamental problem in the field of online algorithms, in which one maintains a cache of $k$ slots as requests for fetching pages arrive online. In the weighted variant of this problem, each page has its own fetching cost; a substantial line of work on this problem culminated in an (optimal) $O(\log k)$-competitive randomized algorithm, due to Bansal, Buchbinder and Naor (FOCS'07).

Existing work for weighted paging assumes that page weights are known in advance, which is not always the case in practice. For example, in multi-level caching architectures, the expected cost of fetching a memory block is a function of its probability of being in a mid-level cache rather than the main memory. This complex property cannot be predicted in advance; over time, however, one may glean information about page weights through sampling their fetching cost multiple times. We present the first algorithm for online weighted paging that does not know page weights in advance, but rather learns from weight samples. In terms of techniques, this requires providing (integral) samples to a fractional solver, requiring a delicate interface between this solver and the randomized rounding scheme; we believe that our work can inspire online algorithms to other problems that involve cost sampling.

## 1 Introduction

**Online weighted paging.** In the online weighted paging problem, or OWP, one is given a cache of $k$ slots, and requests for pages arrive online. Upon each requested page, the algorithm must ensure that the page is in the cache, possibly evicting existing pages in the process. Each page $p$ also has a weight $w_p$, which represents the cost of fetching the page into the cache; the goal of the algorithm is to minimize the total cost of fetching pages. Assuming that the page weights are known, this problem admits an $O(\log k)$-competitive randomized online algorithm, due to Bansal, Buchbinder, and Naor [2010, 2012]; This is optimal, as there exists an $\Omega(\log k)$-competitiveness lower bound for randomized algorithms due to Fiat et al. [1991] (that holds even for the unweighted case).

However, all previous work on paging assumes that the page weights are known *in advance*. This assumption is not always justified; for example, the following scenario, reminiscent of real-world architectures, naturally gives rise to unknown page weights. Consider a multi-core architecture, in which data can be stored in one of the following: a local "L1" cache, unique to each core; a global "L2" cache, shared between the cores; and the (large but slow) main memory. As a specific core requests memory blocks, managing its L1 cache can be seen as an OWP instance. Suppose the costs

---

[*]Research conducted while the author was an intern at Amazon Science.
[†]Research conducted while the author was at Amazon Science.

of fetching a block from the main memory and from the L2 cache are 1 and $\epsilon \ll 1$, respectively. Then, when a core demands a memory block, the expected cost of fetching this block (i.e., its weight) is a convex combination of 1 and $\epsilon$, weighted by the probability that the block is in the L2 cache; this probability can be interpreted as the demand for this block by the various cores. When managing the L1 cache of a core, we would prefer to evict blocks with low expected fetching cost, as they are more likely to be available in the L2 cache. But, this expected cost is a complicated property of the computation run by the cores, and estimating it in advance is infeasible; however, when a block is fetched in the above example, we observe a stochastic cost of either 1 or $\epsilon$. As we sample a given block multiple times, we can gain insight into its weight.

**Multi-armed bandit.** The above example, in which we learn about various options through sampling, is reminiscent of the multi-armed bandit problem, or MAB. In the cost-minimization version of this problem, one is given $n$ options (or arms), each with its own cost in $[0, 1]$. At each time step, the algorithm must choose an option and pay the corresponding cost; when choosing an option $p$, rather than learning its cost $w_p$, the algorithm is only revealed a sample from some distribution whose expectation is $w_p$. In this problem, the goal is to minimize the **regret**, which is the difference between the algorithm's total cost and the optimal cost (which is to always choose the cheapest option). Over $T$ time steps, the best known regret bound for this problem is $\tilde{O}(\sqrt{nT})$, achieved through multiple techniques. (See, e.g., Slivkins et al. [2019], Lattimore and Szepesvári [2020]).

## 1.1 Our Results

We make the first consideration of OWP where page weights are not known in advance, and show that the optimal competitive ratio of $O(\log k)$ can still be obtained. Specifically, we present the problem of OWP-UW (Online Weighted Paging with Unknown Weights), that combines OWP with bandit-like feedback. In OWP-UW, every page $p$ has an arbitrary distribution, whose expectation is its weight $0 < w_p \leq 1$. Upon fetching a page, the algorithm observes a random, independent sample from the distribution of the page. We present the following theorem for OWP-UW.

**Theorem 1.1.** *There exists a randomized algorithm* ON *for* OWP-UW *such that, for every input $Q$,*

$$\mathbb{E}[\text{ON}(Q)] \leq O(\log k) \cdot \text{OPT}(Q) + \tilde{O}(\sqrt{nT}),$$

*where* ON$(Q)$ *is the cost of* ON *on $Q$,* OPT$(Q)$ *is the cost of the optimal solution to $Q$, and the expectation is taken over both the randomness in* ON *and the samples from the distributions of pages.*

Note that the bound in Theorem 1.1 combines a competitive ratio of $O(\log k)$ with a regret (i.e., additive) term of $\tilde{O}(\sqrt{nT})$. To motivate this type of bound, we observe that OWP-UW does not admit sublinear regret without a competitive ratio. Consider the lower bound of $\Omega(\log k)$ for the competitive ratio of paging; stated simply, one of $k + 1$ pages of weight 1 is requested at random. Over a sequence of $T$ requests, the expected cost of any online algorithm is $\Omega(T/k)$; meanwhile, the expected cost of the optimal solution is at most $O(T/(k \log k))$. (The optimal solution would be to wait for a maximal phase of requests containing at most $k$ pages, whose expected length is $\Theta(k \log k)$, then change state at constant cost.) Without a competitive ratio term, the difference between the online and offline solutions is $\Omega(T/k)$, i.e., linear regret. We note that this kind of bound appears in several previous works such as Basu et al. [2019], Foussoul et al. [2023]. As OWP-UW generalizes both standard OWP and MAB, both the competitive ratio and regret terms are asymptotically tight: a competitiveness lower bound of $\Omega(\log k)$ is known for randomized algorithms for online (weighted) paging [Fiat et al., 1991], and a regret lower bound of $\tilde{\Omega}(\sqrt{nT})$ is known for MAB [Lattimore and Szepesvári, 2020].

## 1.2 Our Techniques

**Interface between fractional solution and rounding scheme.** Randomized online algorithms are often built of the following components:

1. A deterministic, $\alpha$-competitive online algorithm for a fractional relaxation of the problem.
2. An online randomized rounding scheme that encapsulates any online fractional algorithm, and has expected cost $\beta$ times the fractional cost.

Combining these components yields an $\alpha\beta$-competitive randomized online (integral) algorithm.

For our problem, it is easy to see where this common scheme fails. The fractional algorithm cannot be competitive without sampling pages; but, pages are sampled by the rounding scheme! Thus, the competitiveness of the fractional algorithm is not independent of the randomized rounding, which must provide samples. One could think of addressing this by feeding any samples obtained by the rounding procedure into the fractional algorithm. However, as the rounding is randomized, this would result in a non-deterministic fractional algorithm. As described later in the paper, this is problematic: the rounding scheme demands a globally accepted fractional solution against which probabilities of cache states are balanced.

Instead, we outline a sampling interface between the fractional solver and the rounding scheme. Once the total fractional eviction of a page reaches an integer, the fractional algorithm will pop a sample of the page from a designated sampling queue, and process that sample. On the other side of the interface, the rounding scheme fills the sampling queue and ensures that when the fractional algorithm demands a sample, the queue will be non-empty with probability 1.

**Optimistic fractional algorithm, pessimistic rounding scheme.** When learning from samples, one must balance the exploration of unfamiliar options and the exploitation of familiar options that are known to be good. A well-known paradigm for achieving this balance in multi-armed bandit problems is *optimism under uncertainty*. Using this paradigm to minimize total cost, one maintains a lower confidence bound (LCB) for the cost of an option, which holds with high probability, and tightens upon receiving samples; then, the option with the lowest LCB is chosen. As a result, one of the following two cases holds: either the option was good (high exploitation); or, the option was bad, which means that the LCB was not tight, and henceforth sampling greatly improves it (high exploration).

Our fractional algorithm for weighted paging employs this method. It optimistically assumes that the price of moving a page is cheap, i.e., is equal to some lower confidence bound (LCB) for that page. It then uses multiplicative updates to allocate servers according to these LCB costs. The optimism under uncertainty paradigm then implies that the fractional algorithm learns the weights over time.

However, the rounding scheme behaves very differently. Unlike the fractional algorithm, the (randomized) rounding scheme is not allowed to use samples to update the confidence bounds; otherwise, our fractional solution would behave non-deterministically. Instead, the rounding scheme takes a pessimistic view: it uses an *upper* confidence bound (UCB) as the cost of a page, thus assuming that the page is expensive. Such pessimistic approaches are common in scenarios where obtaining additional samples is not possible (e.g., offline reinforcement learning [Levine et al., 2020]), but rarely appear as a component of an online algorithm as we suggest in this paper.

## 1.3   Related Work

The online paging problem is a fundamental problem in the field of online algorithms. In the unweighted setting, the optimal competitive ratio for a deterministic algorithm is $k$, due to Sleator and Tarjan [1985]. Allowing randomization improves the best possible competitive ratio to $\Theta(\log k)$ [Fiat et al., 1991]. As part of a line of work on weighted paging and its variants (e.g., Young [1994], Manasse et al. [1990], Albers [2003], Irani [2002], Fiat and Mendel [2000], Bansal et al. [2008], Irani [1997]), the best competitive ratios for weighted paging were settled, and were seen to match the unweighted setting: $k$-competitiveness for deterministic algorithms, due to Chrobak et al. [1991]; and $\Theta(\log k)$-competitiveness for randomized algorithms, due to Bansal et al. [2012].

Online (weighted) paging is a special case of the $k$-server problem, in which $k$ servers exist in a general metric space, and must be moved to address requests on various points in this space; the cache slots in (weighted) paging can be seen as servers, moving in a (weighted) uniform metric space. The $\Theta(k)$ bound on optimal competitiveness in the deterministic for paging also extends to general $k$-server [Manasse et al., 1990, Koutsoupias and Papadimitriou, 1995]. However, allowing randomization, a recent breakthrough result by Bubeck et al. [2023] was a lower bound of $\Omega(\log^2 k)$-competitiveness for $k$-server, diverging from the $O(\log k)$-competitiveness possible for paging.

Multi-Armed Bandit (MAB) is one of the most fundamental problems in online sequential decision making, often used to describe a trade-off between exploration and exploitation. It was extensively studied in the past few decades, giving rise to several algorithmic approaches that guarantee optimal regret. The most popular methods include Optimism Under Uncertainty (e.g., the UCB algorithm [Lai and Robbins, 1985, Auer et al., 2002a]), Action Elimination [Even-Dar et al., 2006], Thompson

Sampling [Thompson, 1933, Agrawal and Goyal, 2012] and Exponential Weights (e.g., the EXP3 algorithm [Auer et al., 2002b]). For a comprehensive review of the MAB literature, see Slivkins et al. [2019], Lattimore and Szepesvári [2020].

## 2   Preliminaries

In OWP-UW, we are given a memory cache of $k$ slots. A sequence of $T$ page requests then arrives in an online fashion; we denote the set of requested pages by $P$, define $n := |P|$, and assume that $n > k$. Each page $p$ has a corresponding weight $0 < w_p \le 1$; the weights are not known to the algorithm. Moreover, every page $p$ has a distribution $\mathcal{D}_p$ supported in $(0, 1]$, such that $\mathbb{E}_{x \sim \mathcal{D}_p}[x] = w_p$.

The online scenario proceeds in $T$ rounds.[3] In each round $t \in \{1, 2, \ldots, T\}$:

- A page $p_t \in P$ is requested.

- If the requested page is already in the cache, then it is immediately served.

- Otherwise, we experience a cache miss, and we must fetch $p_t$ into the cache; if the cache is full, the algorithm must evict some page from the cache to make room for $p_t$.

- Upon evicting any page $p$ from the cache, the algorithm receives an independent sample from $\mathcal{D}_p$.

The algorithm incurs cost when evicting pages from the cache: when evicting a page $p$, the algorithm incurs a cost of $w_p$.[4] Our goal is to minimize the algorithm's total cost of evicting pages, denoted by ON, and we measure our performance by comparison to the total cost of the optimal algorithm, denoted by OPT. We say that our algorithm is $\alpha$-competitive with $\mathcal{R}$ regret if $\mathbb{E}[\text{ON}] \le \alpha \cdot \text{OPT} + \mathcal{R}$.

## 3   Algorithmic Framework and Analysis Overview

We present an overview of the concepts and algorithmic components we use to address OWP-UW. We would like to follow the paradigm of solving a fractional problem online, and then randomly rounding the resulting solution; however, as discussed in the introduction, employing this paradigm for OWP-UW requires a well-defined interface between the fractional solver and the rounding procedure. Thus, we present a fractional version of OWP-UW that captures this interface.

**Fractional OWP-UW.** In fractional OWP-UW, one is allowed to move fractions of servers, and a request for a page is satisfied if the total server fraction at that point sums to 1. More formally, for every page $p \in P$ we maintain an amount $y_p \in [0, 1]$ which is the fraction of $p$ *missing* from the cache; we call $y_p$ the fractional **anti-server** at $p$. (The term anti-server comes from the related $k$-server problem.) The feasibility constraints are:

1. At any point in the algorithm, it holds that $\sum_{p \in P} y_p \ge n - k$. (I.e., the total number of pages in the cache is at most $k$.)

2. After a page $p$ is requested, it holds that $y_p = 0$. (I.e., there exists a total server fraction of 1 at $p$.)

Evicting an $\epsilon$ server fraction from $p$ (i.e., increasing $y_p$ by $\epsilon$) costs $\epsilon \cdot w_p$.

*Sampling.* The fractional algorithm must receive samples of pages over time in order to learn about their weights. An algorithm for fractional OWP-UW receives a sample of a page $p$ whenever the total fraction of $p$ evicted by the algorithm reaches an *integer*. In particular, the algorithm obtains the first sample of $p$ (corresponding to 0 eviction) when $p$ is first requested in the online input.

---

[3] We make a simplifying assumption that $T$ is known in advanced. This can be easily removed using a standard doubling (see, e.g., Slivkins et al. [2019]).

[4] Note that charging an OWP solution for evicting rather than fetching pages is standard; indeed, with the exception of at most $k$ pages, every fetched page is subsequently evicted, and thus the difference between eviction and fetching costs is at most $k$. Moreover, as we analyze additive regret, note that $k \le \sqrt{nT}$, implying that using fetching costs would not affect the bounds in this paper. Finally, note that we sample upon eviction rather than upon fetching, which is the "harder" model.

**Algorithmic components.** We present the fractional algorithm and randomized rounding scheme.

*Fractional algorithm.* In Section 4, we present an algorithm ONF for fractional OWP-UW. Fixing the random samples from the pages' weight distributions, the fractional algorithm ONF is deterministic. For every page $p \in P$, the fractional algorithm maintains an upper confidence bound $\mathrm{UCB}_p$ and a lower confidence bound $\mathrm{LCB}_p$. These confidence bounds depend on the samples provided for that page; we define the *good event* $\mathcal{E}$ to be the event that at every time and for every page $p \in P$, it holds that $\mathrm{LCB}_p \leq w_p \leq \mathrm{UCB}_p$. We later show that $\mathcal{E}$ happens with high probability, and analyze the complementary event separately[5]. Thus, we henceforth focus on the good event.

The following lemma bounds the cost of ONF subject to the good event. In fact, it states a stronger bound, that applies also when the cost of evicting page $p$ is the upper confidence bound $\mathrm{UCB}_p \geq w_p$.

**Lemma 3.1.** *Fixing any input $Q$ for fractional* OWP-UW*, and assuming the good event, it holds that*

$$\mathrm{ONF}(Q) \leq \overline{\mathrm{ONF}}(Q) \leq O(\log k) \cdot \mathrm{OPT}(Q) + \tilde{O}(\sqrt{nT})$$

*where $\overline{\mathrm{ONF}}$ is the cost of the algorithm on the input where the cost of evicting a page $p$ is $\mathrm{UCB}_p \geq w_p$.*

*Randomized rounding.* In Section 5 we present the randomized algorithm ON for (integral) OWP-UW. It maintains a probability distribution over integral cache states by holding an instance of ONF, to which it feeds the online input. For the online input to constitute a valid *fractional* input, the randomized algorithm ensures that samples are provided to ONF when required. In addition, the randomized algorithm makes use of ONF's exploration of page weights; specifically, it uses the UCBs calculated by ONF.

**Lemma 3.2.** *Fixing any input $Q$ for (integral)* OWP-UW*, assuming the good event $\mathcal{E}$, it holds that*

$$\mathbb{E}[\mathrm{ON}(Q)] \leq O(1) \cdot \overline{\mathrm{ONF}}(Q) + n$$

*where $\overline{\mathrm{ONF}}(Q)$ is the cost of the algorithm on $Q$ such that the cost of evicting a page $p$ is $\mathrm{UCB}_p \geq w_p$.*

Figure 1 provides a step-by-step visualization of the interface between the fractional algorithm and the rounding scheme over the handling of a page request.

## 4 Algorithm for Fractional OWP-UW

We now describe our algorithm for the fractional relaxation of OWP-UW, proving Lemma 3.1. Our fractional algorithm, presented in Algorithm 1 below, uses samples provided by the rounding scheme to learn the weights. A new sample for page $p$ is provided and processed whenever the sum of fractional movements (in absolute value) $m_p$ hits a natural number. (At this point the number of samples $n_p$ is incremented.) The algorithm calculates non-increasing UCBs and non-decreasing LCBs that will be specified later in Section C and guarantee with high probability, for every page $p \in P$ and time step $t \in [1, T]$, $\mathrm{LCB}_p \leq w_p \leq \mathrm{UCB}_p$.

At each time step $t$, upon a new page request $p_t$, the algorithm updates its feasible fractional cache solution $\{y_p\}_{p \in P}$. The fractions are computed using optimistic estimates of the weights, i.e., the LCBs, in order to induce exploration and allow the true weights to be learned over time. After serving page $p_t$ (that is, setting $y_{p_t} = 0$), the algorithm continuously increases the anti-servers of all the other pages in the cache until feasibility is reached (that is, until $\sum_{p \in P} y_p = n - k$). The fraction $y_p$ for some page $p$ in the cache is increased proportionally to $\frac{y_p + \eta}{\mathrm{LCB}_p}$, which is our adaption of the algorithmic approach of Bansal et al. [2010] to the unknown-weights scenario. Finally, to fulfil its end in the interface, the fractional algorithm passes its feasible fractional solution to the rounding scheme together with pessimistic estimates of the weights, i.e, the UCBs.

### 4.1 Analysis

In this analysis section, our goal is to bound the amount $\overline{\mathrm{ONF}}$ with respect to the UCBs and LCBs calculated by the algorithm; i.e., to prove Lemma 4.1. Lemma C.1 and Lemma C.2 from Appendix C then make the choice of confidence bounds concrete, such that combining it with Lemma 4.1 yields the final bound for the fractional algorithm, i.e., Lemma 3.1.

---

[5]Specifically, we show that the complementary event $\overline{\mathcal{E}}$ happens with probability at most $\frac{1}{nT}$, and that the algorithm's cost is at most $nT$ times the optimal cost when it happens.

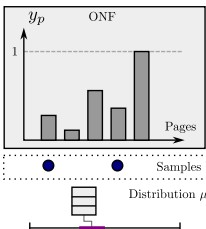 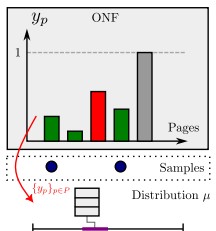 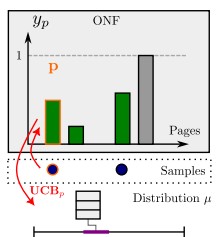 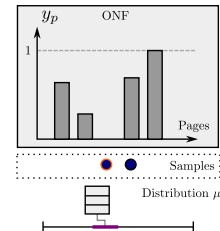

This figure visualizes the running of the algorithm over a request in the input. (a) shows the state prior to the arrival of the request. The integral algorithm maintains an instance of the fractional algorithm ONF, as well as a distribution over integral cache states that upholds some properties w.r.t. ONF (specifically, the consistency property and the subset property); note that these properties are a function of both the anti-server state $\{y_p\}_{p \in P}$ and the upper-confidence bounds $\{\text{UCB}_p\}_{p \in P}$ in ONF. The integral algorithm also maintains a set of page samples, to be demanded by ONF at a later time. In (b), a page is requested (in red); thus, the fractional algorithm must fetch it into the cache, i.e., set its anti-server to 0. To maintain feasibility, the fractional algorithm will increase the anti-server at other pages in which some server fraction exists (in green). These changes in anti-server are also fed into the integral algorithm, which modifies its distribution to maintain consistency and the subset property w.r.t. ONF. In (c), ONF reaches integral total eviction of a page $p$, and demands a sample from the integral algorithm. (We show that such a sample always exists when demanded.) The bound $\text{UCB}_p$ is updated in ONF, and is then fed to the integral algorithm to maintain the desired properties. After this sample, continuous increasing of anti-server continues until feasibility is reached in (d). Then, the integral algorithm ensures that a sample exists for the requested page $p_t$, sampling the page if needed. (As $p_t$ now exists in $\mu$ with probability 1, sampling is done through evicting and re-fetching $p_t$.)

Figure 1: Visualization of the interface between the fractional and integral algorithms

---

**Algorithm 1:** Fractional Online Weighted Caching with Unknown Weights

1   Set $\eta \leftarrow 1/k$ and $y_p \leftarrow 1$ for every $p \in P$.
2   **for** *time $t = 1, 2, ..., T$* **do**
3      Page $p_t \in P$ is requested.
4      **continually increase** $y_p, m_p$ *in proportion to* $\frac{y_p + \eta}{\text{LCB}_p}$ *for every* $p \in P \setminus \{p_t\}$ *where* $y_p < 1$ **until:**
5          **if** *$m_p$ reaches an integer for some $p \in P \setminus \{p_t\}$* **then**
6              **receive sample** $\widetilde{w}_p$ for $p$.
7              set $n_p \leftarrow n_p + 1$.
8              call UPDATECONFBOUNDS$(p, \widetilde{w}_p)$. *// recalculate confidence bounds for $p$*
9          **if** $\sum_{p \in P} y_p = n - k$ **then break** from the continuous increase loop.
10      **if** *this is the first request for $p_t$* **then**
11          **receive sample** $\widetilde{w}_{p_t}$ for $p_t$.
12          define $m_p \leftarrow 0, n_p \leftarrow 1$.
13          call UPDATECONFBOUNDS$(p_t, \widetilde{w}_{p_t})$. *// calculate confidence bounds for $p_t$*

---

**Lemma 4.1.** *Fixing any input $Q$ for fractional* OWP-UW, *and assuming the good event, it holds that*

$$\overline{\text{ONF}}(Q) \leq O(\log k) \cdot \text{OPT}(Q) + \sum_{p \in P} \sum_{i=1}^{n_p} \left(\text{UCB}_{p,i} - \text{LCB}_{p,i}\right) + 2\log(1 + 1/\eta) \sum_{p \in P} \text{LCB}_p.$$

*where **(a)** $\overline{\text{ONF}}$ is the cost of the algorithm on the input such that the cost of evicting a page $p$ is $\text{UCB}_p \geq w_p$, and **(b)** $\text{UCB}_{p,i}, \text{LCB}_{p,i}$ are the values of $\text{UCB}_p$ and $\text{LCB}_p$ calculated by the procedure UPDATECONFBOUNDS (found in Appendix C) immediately after processing the $i$'th sample of $p$, and **(c)** $\text{LCB}_p$ is the value after the last sample of page $p$ was processed.*

*Proof sketch.* We prove the lemma using a potential analysis. In Bansal et al. [2010], a potential function was introduced that encodes the discrepancy between the state of the optimal solution and the state of the algorithm. In our case, we require an additional term which can be viewed as a fractional exploration budget. This budget is "recharged" upon receiving a sample; the cost of this recharging goes into a regret term. □

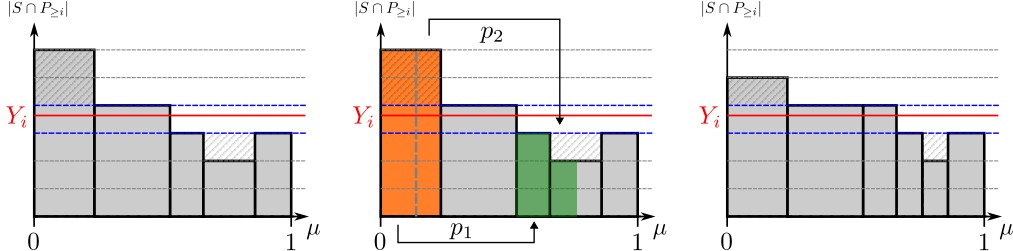

This figure visualizes a single step in REBALANCESUBSETS. Subfigure (a) shows the distribution of anti-cache states prior to this step; specifically, the $x$ axis is the probability measure, and the $y$-axis is the number of pages of class $i$ and above in the anti-cache, i.e., $m := |S \cap P_{\geq i}|$. The red line is $Y_i$, which through consistency, is the expectation of $m$; the blue dotted lines are thus the allowed values for $m$, which are $\lceil\lceil Y_i \rceil, \lfloor Y_i \rfloor\rceil$. The total striped area in the figure is the imbalance measure, formally defined in Definition B.2. Subfigure (b) shows a single rebalancing step; we choose the imbalanced anti-cache $S$ that maximizes $|m - Y_i|$; in our case, $m > Y_i$, and thus we match its measure with an identical measure of anti-cache states that are below the upper blue line, i.e., can receive a page without increasing imbalance. Then, a page of class $i$ is handed from $S$ to every matched state $S'$; note that every matched state might get a different page from $S$, but some such page in $S \setminus S'$ is proven to exist. Finally, Subfigure (c) shows the state after the page transfer; note the decrease in imbalance that results. The REBALANCESUBSETS procedure performs such steps until there is no imbalance; then, the procedure would advance to class $i - 1$.

Figure 2: Example of a rebalancing step

## 5 Randomized Rounding

This section describes a randomized algorithm for (integral) OWP-UW, which uses Algorithm 1 for fractional OWP-UW to maintain a probability distribution over valid integral cache states, while obtaining and providing page weight samples to Algorithm 1. The method in which the randomized algorithm encapsulates and tracks the fractional solution is inspired by Bansal et al. [2012], which maintains a balanced property over weight classes of pages. However, as the weights are unknown in our case, the classes are instead defined using the probabilistic bounds maintained by the fractional solution (i.e., the UCBs). But, these bounds are dynamic, and change over the course of the algorithm; the imbalance caused by these discrete changes increases exponentially during rebalancing, and thus requires a more robust rebalancing procedure.

Following the notation in the previous sections, we identify each cache state with the set of pages *not* in the cache. Observing the state of the randomized algorithm at some point in time, let $\mu(S)$ be the probability that $S \subseteq P$ is the set of pages missing from the cache, also called the *anti-cache*. For the algorithm to be a valid algorithm for OWP-UW, the cache can never contain more than $k$ pages; this is formalized in the following property.

**Definition 5.1** (valid distribution). A probability distribution $\mu$ is valid, if for any set $S \subseteq P$ with $\mu(S) > 0$ it holds that $|S| \geq n - k$.

Instead of maintaining the distribution's validity, we will maintain a stronger property that implies validity. This property is the *balanced* property, involving the UCBs calculated by the fractional algorithm.

For every page $p$, we define the $i$'th UCB class to be $P_i := \{p \in P : 6^i \leq \text{UCB}_p < 6^{i+1}\}$. (Note that $\text{UCB}_p \in (0, 1]$.) We also define $P_{\geq j} := \bigcup_{i \geq j} P_j$, the set of all pages that their UCB is at least $6^j$.

Let $\{y_p\}_{p \in P}$ be the fractional solution. The balanced property requires that, for every set $S$ such that $\mu(S) > 0$ and every index $j$, the number of pages in $S$ of class at least $j$ is the same as in the fractional solution, up to rounding. Formally, we define the balanced property as follows.

**Definition 5.2** (balanced distribution). A probability distribution $\mu$ has the balanced subsets property with respect to $y$, if for any set $S \subseteq P$ with $\mu(S) > 0$, the following holds for all $j$:

$$\left\lfloor \sum_{i \geq j} \sum_{p \in P_i} y_p \right\rfloor \leq \sum_{i \geq j} \sum_{p \in P_i} \mathbb{I}[p \in S] \leq \left\lceil \sum_{i \geq j} \sum_{p \in P_i} y_p \right\rceil.$$

---

**Algorithm 2:** Randomized rounding algorithm for OWP-UW

---

1 **Initialization**
2      Let ONF be an instance of Algorithm 1, that maintains a fractional anti-server allocation $\{y_p\}_{p \in P}$.
3      Define $\mu$ to be a distribution over cache states, initially containing the empty cache state with
         probability 1.
4      For every $p \in P$, let $s_p \leftarrow$ NULL.

5 **Event Function** UPONREQUEST($p$) *// called upon a request for page p*
6      pass the request for $p$ to ONF.
7      **while** ONF *is handling the request for p* **do** *// loop of Line 4 in Alg. 1*
8          **if** ONF *increases $y_{p'}$ by $\epsilon$, for some $p' \in P$* **then**
9              add $p'$ to the anti-cache in an $\epsilon$-measure of states without $p'$.
10              call REBALANCESUBSETS.
11          **if** ONF *decreases $y_{p'}$ by $\epsilon$, for some $p' \in P$* **then**
12              remove $p'$ from the anti-cache in an $\epsilon$-measure of states with $p'$.
13              call REBALANCESUBSETS.
14          **if** ONF *samples a page $p' \in P$* **then** *// sample due to Line 6 of Alg. 1*
15              provide $s_{p'}$ as a sample to ONF, and set $s_{p'} \leftarrow$ NULL.
16              call REBALANCESUBSETS. *// rebalance due to change in UCB$_{p'}$.*
17      **if** $s_p =$ NULL **then** evict and re-fetch $p$ to obtain weight sample $\tilde{w}_p$, and set $s_p \leftarrow \tilde{w}_p$.
18      **if** ONF *requests a sample of p* **then** *// sample due to Line 11 of Alg. 1*
19          provide $s_p$ as a sample to ONF, and set $s_p \leftarrow$ NULL.
20          call REBALANCESUBSETS. *// rebalance due to change in UCB$_p$.*

---

---

**Algorithm 3:** Rebalancing procedure for randomized algorithm

---

1 **Function** REBALANCESUBSETS
2      let $j_{\max}$ be the maximum class that is not balanced.
3      let $j_{\min} := \lceil \log(\text{UCB}_{\min}) \rceil$, where $\text{UCB}_{\min} := \min_{p \in P} \text{UCB}_p$.
4      for every class $j$, let $P_j := \{p \in P \,\big|\, \lceil \log(\text{UCB}_P) \rceil = j\}$.
5      **for** $j$ *from $j_{\max}$ down to $j_{\min}$* **do**
6          let $P_{\geq j} := \bigcup_{j' \geq j} P_{j'}$.
7          let $Y_j := \sum_{p \in P_{\geq j}} y_p$.
8          **while** $\exists S$ *s.t.* $\mu(S) > 0$ *and* $|S \cap P_{\geq j}| \notin \{\lceil Y_j \rceil, \lfloor Y_j \rfloor\}$ **do** *// iteratively eliminate imbalanced states*
9              choose such $S$ that maximizes $|m - Y_j|$, where $m := |S \cap P_{\geq j}|$.
10              **if** $m \geq \lceil Y_j \rceil + 1$ **then**
11                  Match the $\mu(S)$ measure of $S$ with an identical measure of anti-cache states with at most
                     $\lceil Y_j \rceil - 1$ pages from $P_{\geq j}$.
12                  **foreach** *anti-cache state $S'$ matched with $S$ at measure $x \leq \mu(S)$* **do**
13                      identify a page $p \in P_j$ such that $p \in S \setminus S'$.
14                      remove $p$ from the anti-cache in the $x$ measure of $S$, and insert it into the anti-cache in
                       the $x$ measure of $S'$.
15              **if** $m \leq \lfloor Y_j \rfloor - 1$ **then**
16                  Match the $\mu(S)$ measure of $S$ with an identical measure of anti-cache states with at least
                     $\lfloor Y_j \rfloor + 1$ pages from $P_{\geq j}$.
17                  **foreach** *anti-cache state $S'$ matched with $S$ at measure $x_{S'} \leq \mu(S)$* **do**
18                      identify a page $p \in P_j$ such that $p \in S' \setminus S$.
19                      remove $p$ from the anti-cache in the $x$ measure of $S'$, and insert it into the anti-cache
                     in the $x$ measure of $S$.

---

Choosing the minimum UCB class in Definition 5.2, and noting that $\sum_{p \in P} y_p \geq n - k$ through feasibility, immediately yields the following remark.

*Remark* 5.3. Every balanced probability distribution is also a valid distribution.

To follow the fractional solution, we also demand that the distribution $\mu$ is *consistent* with the fractional solution, meaning, the marginal probability in $\mu$ that any page $p$ is missing from the cache must be equal to $y_p$.

**Definition 5.4** (consistent distribution). A probability distribution $\mu$ on subsets $S \subseteq P$ is consistent with respect to a fractional solution $\{y_p\}_{p \in P}$ if for every page $p$ it holds that $\sum_{S \subseteq P | p \in S} \mu(S) = y_p$.

In the following we describe the online maintenance of the distribution $\mu$, that yields a distribution satisfying all of the above.

**Algorithm overview.** The randomized algorithm for OWP-UW is given in Algorithm 2. The algorithm encapsulates an instance of Algorithm 1 for fractional OWP-UW, called ONF. Upon a new page request $p_t$ at round $t$, the algorithm forwards this requests to ONF. As ONF makes changes to its fractional solution, the algorithm modifies its probability distribution accordingly to remain consistent and balanced (and hence also valid).

Upon any (infinitesimally small) change to a fractional variable, the algorithm first changes its distribution to maintain consistency: when the fractional algorithm ONF increases the variable $y_{p'}$ of any page $p'$ by an $\epsilon$-measure, the algorithm identifies an $\epsilon$-measure of cache states $S \subseteq P$ in which there is no anti-server at $p'$ and adds anti-server at $p'$. The case in which the fractional algorithm decreases a variable is analogous.

However, this procedure may invalidate the balanced property. Specifically, for some class $j$, letting $Y_j$ be the total anti-server fraction of pages of at least class $j$ in ONF, there might now be states with $\lceil Y_j \rceil + 1$ or $\lfloor Y_j \rfloor - 1$ such pages in the anti-cache. Thus, the algorithm makes a call to REBALANCESUBSETS, which restores the balanced property class-by-class, in a descending order. For every class $j$, the procedure repeatedly identifies a violating state $S$ where the number of pages of class $\geq j$ in the anti-cache is not in $\{\lceil Y_j \rceil, \lfloor Y_j \rfloor\}$; suppose it identifies such a state with more than $\lceil Y_j \rceil$ such pages (the case of less than $\lfloor Y_j \rfloor$ pages is analogous). The procedure seeks to move a page of class $j$ from this state to another state in a way that does not increase the "imbalance" in class $j$. Thus, the procedure identifies a matching measure of anti-cache states that contain at most $\lceil Y_j \rceil - 1$ such pages, and moves a page of class $j$ from $S$ to $S'$, for every $S'$ in the matched measure; a visualization of the procedure is given in Figure 2. (The existence of this matching measure, as well as a page to move, are shown in the analysis.) In particular, note that the probability of every page being in the anti-cache remains the same, and thus REBALANCESUBSETS does not impact consistency.

Regarding samples, the algorithm can maintain a sample $s_p$ for every page $p$. A sample for $p$ is obtained upon a request for $p$ after $p$ is fetched with probability 1 into the cache, if no such sample already exists (i.e., $s_p = \text{NULL}$). Whenever ONF requests a sample for a page $p$, the randomized algorithm provides the sample $s_p$, and sets the variable $s_p$ to be NULL (we show that $s_p$ is never NULL when ONF samples $p$). A fine point is the sampling of a new page in Line 11 of Algorithm 1; this happens after Line 17 of Algorithm 2.

# 6 Conclusions

In this paper, we presented the first algorithm for online weighted paging in which page weights are not known in advance, but are instead sampled stochastically. In this model, we were able to recreate the best possible bounds for the classic online problem, with an added regret term typical to the multi-armed bandit setting. This unknown-costs relaxation makes sense because the problem has recurring costs; that is, the cost of evicting a page $p$ can be incurred multiple times across the lifetime of the algorithm, and thus benefits from sampling.

We believe this paper can inspire future work on this problem. For example, revisiting the motivating case of managing a core-local L1 cache, the popularity of a page among the cores can vary over time; this would correspond to the problem of non-stationary bandits (see, e.g., Auer et al. [2019b,a], Chen et al. [2019]), and it would be interesting to apply techniques from this domain to OWP-UW.

Finally, we hope that the techniques outlined in this paper could be extended to additional such problems. Specifically, we believe that the paradigm of using optimistic confidence bounds in lieu of actual costs could be used to adapt classical online algorithms to the unknown-costs setting. In addition, the interface between the fractional solver and rounding scheme could be used to mediate integral samples to an online fractional solver, which is a common component in many online algorithms.

## Acknowledgments

This project has received funding from the European Research Council (ERC) under the European Union's Horizon 2020 research and innovation program (grant agreement No. 882396), by the Israel Science Foundation, the Yandex Initiative for Machine Learning at Tel Aviv University and a grant from the Tel Aviv University Center for AI and Data Science (TAD).

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

# A  Analysis of the Fractional Algorithm

In this analysis section, our goal is to bound the amount $\overline{\mathrm{ONF}}$ with respect to the UCBs and LCBs calculated by the algorithm; i.e., to prove Lemma 4.1. Lemma C.1 and Lemma C.2 from Appendix C then make the choice of confidence bounds concrete, such that combining it with Lemma 4.1 yields the final bound for the fractional algorithm, i.e., Lemma 3.1.

*Proof of Lemma 4.1.* For the sake of this lemma, we assume without loss of generality that the optimal solution is lazy; that is, it only evicts a (single) page in order to fetch the currently-requested page. (It is easy to see that any solution can be converted into a lazy solution of lesser or equal cost.) In the following we present a potential analysis to prove that $\overline{\mathrm{ONF}} \leq 2 \log(1 + k) \cdot \mathrm{OPT} + \mathcal{U}_T$, where $\mathcal{U}_T$ is a regret term summed over $T$ time steps that will be defined later. To that end, we show that the following equation holds for every round $t$,

$$\Delta \overline{\mathrm{ONF}}_t + \Delta \Phi_t \leq 2 \log(1 + 1/\eta) \cdot \Delta \mathrm{OPT}_t + \Delta \mathcal{U}_t, \tag{1}$$

where $\Delta X_t$ is the change in $X$ in time $t$ and $\Phi$ is a potential function that we define next.

Let $C_t^*$ denote the set of pages in the offline (optimal) cache at time $t$. The potential function we chose is an adaptation of the potential function used by Bansal et al. [2010], but modified to encode the uncertainty cost for not knowing the true weights. We define it as follows.

$$\Phi_t = -2 \sum_{p \notin C_t^*} \mathrm{LCB}_p \cdot \log\left(\frac{y_p + \eta}{1 + \eta}\right) + \sum_{p \in P} \left(\mathrm{UCB}_p - \mathrm{LCB}_p\right) \cdot \left(n_p - m_p\right),$$

where $\mathrm{UCB}_p, \mathrm{LCB}_p$ are the confidence bounds in time step $t$, $n_p$ is the number of samples of page $p$ collected until that point, and $m_p$ is the total fractional movement of that page until that point. Note that $n_p - m_p \in [0, 1]$. We now show that Equation 1 holds in the three different cases in which the costs of the potential or the regret change. Before moving forward, we define the regret term, denoted $\mathcal{U}$, as follows,

$$\mathcal{U} := \sum_{p \in P} \sum_{i=1}^{n_p} (\mathrm{UCB}_{p,i} - \mathrm{LCB}_{p,i}) + 2 \log(1 + 1/\eta) \sum_{p \in P} \mathrm{LCB}_p.$$

We note that each time the algorithm gets a new sample, $n_p$ is incremented and $\mathcal{U}$ increases.

**Case 1 - the optimal algorithm moves.**  Note that a change in the cache of the optimal solution does not affect $\overline{\mathrm{ONF}}$ or $\mathcal{U}$, and thus $\Delta \overline{\mathrm{ONF}} = \Delta \mathcal{U} = 0$. However, $\Delta \mathrm{OPT}$ might be non-zero, as the optimal solution incurs a moving cost; in addition, $\Delta \Phi$ might be non-zero, as the change in the optimal cache might affect the summands in the first term of the potential function.

Thus, proving Equation (1) for this case reduces to proving $\Delta \Phi_t \leq 2 \log(1 + 1/\eta) \Delta \mathrm{OPT}$. Assume the optimal solution moves; as we assume that the optimal solution is lazy, it must be that the requested page $p_t$ is not in $C_{t-1}^*$, and that the optimal solution fetches it into the cache, possibly evicting a (single) page from its cache.

First, consider the case in which there exists an empty slot in the cache, and no evictions take place when fetching $p_t$. In this case, $\Delta \mathrm{OPT} = 0$; moreover, as $C_{t-1}^* \subseteq C_t^*$, it holds that $\Delta \Phi \leq 0$. Thus, Equation (1) holds.

Otherwise, at time $t$, page $p_t$ is fetched and another page $p$ is evicted. Thus, $\Delta \mathrm{OPT} = w_p$. For the potential change, $p$ starts to contribute to $\Phi$. In the worst case, $y_p = 0$ and then $\Phi$ is increased by at most $2 \mathrm{LCB}_p \log(1 + 1/\eta) \leq 2 w_p \log(1 + 1/\eta) = 2 \log(1 + 1/\eta) \Delta \mathrm{OPT}$, as desired.

**Case 2 - the fractional algorithm moves.**  In this case, only the potential and the cost of the fractional algorithm change, i.e., $\Delta \mathrm{OPT}_t = \Delta \mathcal{U}_t = 0$. Thus, Equation (1) reduces to proving $\Delta \overline{\mathrm{ONF}}_t + \Delta \Phi_t \leq 0$ in this case.

There are two types of movement made by the algorithm: the immediate fetching of the requested page $p_t$, and the continuous eviction of other pages from the cache until feasibility is reached (i.e., there are $n - k$ anti-servers). First, consider the fetching of $p_t$ into the cache; as we charge for

evictions, this action does not incur cost ($\Delta\overline{\text{ONF}} = 0$). In addition, through OPT's feasibility, it holds that $p_t \in C_t^*$, and thus changing $y_{p_t}$ does not affect the potential function ($\Delta\Phi = 0$); thus, Equation (1) holds for this sub-case.

Next, consider the continuous eviction of pages. Suppose the fractional algorithm evicts $d_y$ page units, where $d_y$ is infinitesimally small. Let $S = \{p_t\} \cup \{p : y_p < 1\}$. Since the fractional algorithm increases $y_p$ proportionally to $\frac{y_p+\eta}{\text{LCB}_p}$ for each $p \in S \setminus \{p_t\}$, it holds that, $dy_p = \frac{1}{N} \cdot \frac{y_p+\eta}{\text{LCB}_p} dy$, for $N := \sum_{p \in S \setminus \{p_t\}} \frac{y_p+\eta}{\text{LCB}_p}$. Thus, $\Delta\overline{\text{ONF}}$ can be bounded as follows.

$$
\begin{aligned}
\Delta\overline{\text{ONF}} &\leq \sum_{p \in S \setminus \{p_t\}} \text{UCB}_p dy_p \\
&= \sum_{p \in S \setminus \{p_t\}} \left(\text{LCB}_p + \left(\text{UCB}_p - \text{LCB}_p\right)\right) dy_p \\
&= \sum_{p \in S \setminus \{p_t\}} \left(\text{LCB}_p + \left(\text{UCB}_p - \text{LCB}_p\right)\right) \frac{1}{N} \frac{y_p+\eta}{\text{LCB}_p} dy \\
&= \sum_{p \in S \setminus \{p_t\}} \frac{1}{N}\left(y_p + \eta\right) dy + \sum_{p \in S \setminus \{p_t\}} \left(\text{UCB}_p - \text{LCB}_p\right) dy_p \\
&= \underbrace{\sum_{p \in S \setminus \{p_t\}} \frac{1}{N}\left(y_p + \eta\right) dy}_{(a)} + \underbrace{\sum_{p \in P} \left(\text{UCB}_p - \text{LCB}_p\right) dy_p}_{(b)}.
\end{aligned}
$$

where the final equality stems from having $dy_p = 0$ for every $p \notin S \setminus \{p_t\}$. We now bound the term $(a)$ using similar arguments to those presented in Bansal et al. [2010]. We present them below for completeness.

$$
(a) = \sum_{p \in S \setminus \{p_t\}} \frac{1}{N}\left(y_p + \eta\right) dy \leq \frac{(|S| - k)}{N} dy + \frac{(|S| - 1)\eta}{N} dy \leq \frac{2(|S| - k)}{N} dy,
$$

where the first inequality implied since $y_p = 1$ for $p \notin S$, which yields that

$$
\sum_{p \in S \setminus \{p_t\}} y_p \leq \sum_{p \in S} y_p = \sum_{p \in P} y_p - \sum_{p \notin S} y_p \leq (|P| - k) - (|P| - |S|) = |S| - k,
$$

and the second inequality implied as $(x - 1)\eta \leq x - k$ for any $x \geq k + 1$ and using that $|S| \geq k + 1$.

Next, we upper bound $\Delta\Phi_t$. The rate of change in the potential with respect to $y_p$ is

$$
\frac{d\Phi_t}{dy_p} = -2\frac{\text{LCB}_p}{y_{p,t} + \eta} - \left(\text{UCB}_p - \text{LCB}_p\right).
$$

The non-trivial part in the above calculation is that $\frac{d\left(\text{UCB}_p - \text{LCB}_p\right) \cdot \left(n_p - m_p\right)}{dy_p} = -\left(\text{UCB}_p - \text{LCB}_p\right)$. This is true since $n_p$ is uncorrelated with the change in $y_p$, however, $m_p$ is increasing with with respect to $y_p$ in 1-linear ratio. Using the above we get that

$$
\begin{aligned}
\Delta\Phi_t &= \sum_{p \in P} \left(\frac{d\Phi_t}{dy_p}\right) dy_p \\
&= -2 \sum_{p \in S \setminus C_t^*} \frac{\text{LCB}_p}{y_p + \eta} \frac{1}{N} \frac{y_p + \eta}{\text{LCB}_p} dy - \sum_{p \in P} \left(\text{UCB}_p - \text{LCB}_p\right) \cdot dy_p \\
&= -2 \sum_{p \in S \setminus C_t^*} \frac{dy}{N} - \sum_{p \in P} \left(\text{UCB}_p - \text{LCB}_p\right) \cdot dy_p \\
&\leq -2 \frac{|S| - k}{N} dy - \sum_{p \in P} \left(\text{UCB}_p - \text{LCB}_p\right) \cdot dy_p.
\end{aligned}
$$

Thus, $\Delta\overline{\text{ONF}}_t + \Delta\Phi_t \leq 0$, as required.

**Case 3 - the LCBs are updated (a new sample is processed).** Suppose the algorithm samples a page $p$ for the $i$'th time, and updates $\text{UCB}_p, \text{LCB}_p$ accordingly. Note that this sample does not incur any cost for the fractional algorithm or optimal solution, and thus $\Delta\overline{\text{ONF}} = \Delta\text{OPT} = 0$. Thus, proving Equation (1) reduces to proving $\Delta\Phi \leq \Delta\mathcal{U}$ for this case. Recalling the definition of $\mathcal{U}$, it holds that

$$\Delta\mathcal{U} = (\text{UCB}_{p,i} - \text{LCB}_{p,i}) + 2\log(1 + 1/\eta)(\text{LCB}_{p,i} - \text{LCB}_{p,i-1}).$$

Denote by $n_p, m_p$ the variables of that name prior to sampling $p$; note that $m_p$ remains the same after sampling, while $n_p$ is incremented to $n'_p := n_p + 1$. Moreover, note that sampling always occurs when $m_p = n_p$. Thus, the change in potential function is bounded as follows.

$$\Delta\Phi = 2\log(1 + 1/\eta)\mathbb{I}[p \notin C_i^*](\text{LCB}_{p,i} - \text{LCB}_{p,i-1}) + \text{UCB}_{p,i} - \text{LCB}_{p,i}$$
$$\leq 2\log(1 + 1/\eta)(\text{LCB}_{p,i} - \text{LCB}_{p,i-1}) + \text{UCB}_{p,i} - \text{LCB}_{p,i}$$

where the second inequality is due to the LCBs being monotone non-decreasing. $\qquad\square$

# B  Proofs from Section 5

## B.1  Analysis and Proof of Lemma 3.2

In this subsection, we analyze Algorithm 2 and prove Lemma 3.2. Throughout this subsection, we assume the good event $\mathcal{E}$; that is, the UCBs and LCBs generated by ONF throughout the algorithm are valid upper and lower bounds for the weights of pages.

We start by proving that the algorithm is able to provide a new sample whenever the fractional algorithm requires one.

**Proposition B.1.** *Algorithm 2 provides page weight samples whenever demanded by* ONF.

*Proof.* We must prove that whenever a sample of page $p$ is requested by ONF, it holds that the variable $s_p$ in Algorithm 2 is not NULL. Note that Algorithm 2 samples $s_p$ at the end of a request for $p$. Now, note that:

- The first sample of $p$ is requested after the first request for $p$ is handled by ONF, and thus after $p$ is sampled.

- Between two subsequent requests for samples of $p$ by ONF, the eviction fraction $m_p$ increased by 1. But, this cannot happen without a request for $p$ in the interim; this request ensures that $s_p \neq$ NULL, and thus the second request is satisfied.

Combining both cases, all sample requests by ONF are satisfied by Algorithm 2. $\qquad\square$

Next, we focus on proving that the distribution maintained by the algorithm is consistent and balanced (and hence also valid). To prove this, we first formalize and prove the guarantee provided by the REBALANCESUBSETS procedure. To this end, we define an amount quantifying the degree to which a class is imbalanced in a given distribution.

**Definition B.2.** Let $\mu$ be a distribution over cache states, let $\{y_p\}_{p \in P}$ be the current fractional solution, and let $j$ be some class. We define the *imbalance* of class $j$ in $\mu$ to be

$$\sum_{S \subseteq P} \mu(S) \cdot \max\left\{\left|S \cap P_{\geq j}\right| - \lceil Y_j \rceil, \lfloor Y_j \rfloor - \left|S \cap P_{\geq j}\right|\right\};$$

Definition B.2 quantifies the degree to which a given class is not balanced in a given distribution; one can see that if the class is balanced, this amount would be 0.

**Lemma B.3.** *Suppose* REBALANCESUBSETS *is called, and let $\mu, \mu'$ be the distributions before and after the call to* REBALANCESUBSETS, *respectively; let $j_{\max}$ be the maximum non-balanced class in $\mu$.*

*Suppose that (a) $\mu$ is consistent, and (b) there exists $\epsilon > 0$ such that the imbalance of any class $j \leq j_{\max}$ in $\mu$ is at most $\epsilon$. Then, it holds that $\mu'$ is both consistent and balanced. Moreover, the total cost of* REBALANCESUBSETS *is at most $12\epsilon \cdot 6^{j_{\max}}$.*

*Proof.* The running of REBALANCESUBSETS consists of iterations of the **For** loop in Line 5; we number these iterations according to the class considered in the iteration (e.g., "Iteration $i$" considers class $i$). We make a claim about the state of the anti-cache distribution after each iteration, and prove this claim inductively; applying this claim to the final iteration implies the lemma. Specifically, where $j_{\max}$ as defined in the lemma statement, for every $i \leq j_{\max}$ consider the distribution immediately before Iteration $i$, denoted $\mu_i$. We claim that **(a)** the $\mu_i$ is consistent, (b) all classes greater than $i$ are balanced in $\mu_i$, and (c) classes at most $i$ have imbalance at most $\epsilon \cdot 3^{j_{\max}-i}$ in $\mu_i$. Where $j_{\min}$ is the minimum class, note that $\mu' = \mu_{j_{\min}-1}$, and that this claim implies that $\mu'$ is consistent and balanced. (Note that the claim implies that class $j_{\min}$ is balanced, which also implies that all classes smaller than $j_{\min}$ are balanced.).

We prove this claim by descending induction on $i$. The base case, in which $i = j_{\max}$, is simply a restatement of the assumptions made in the lemma, and thus holds. Now, assume that the claim holds for any class $i \leq j_{\max}$; we now prove it for class $i - 1$.

**Consistency.** First, as we've assumed that $\mu_i$ is consistent, note that Iteration $i$ does not change the marginal probability of a given page $p$ being in the anti-cache, as pages are only moved between identical anti-cache measures. Thus, the anti-cache distribution remains consistent at any step during Iteration $i$; in particular, $\mu_{i-1}$ is consistent.

**Existence of destination measure.** Next, observe that every changes in Iteration $i$ consists of identifying a measure of a violating anti-cache, and matching this measure to an identical measure of anti-cache states to which pages can be moved. To show that the procedure is legal, we claim that this measure always exists. Consider such a change that identifies violating anti-cache $S$, and let $\hat{\mu}$ be the distribution at that point. Assume that $S$ is an "upwards" violation, i.e., $m \geq \lceil Y_i \rceil + 1$, where $m := |S \cap P_{\geq i}|$; the case of a "downwards" violation is analogous. Note that consistency implies that $\mathbb{E}_{S' \sim \hat{\mu}} |S' \cap P_{\geq i}| = Y_i$. Also note that $S$ was chosen to maximize $|m - Y_i|$, i.e., the distance from the expectation. Thus, there exists a measure of at least $\hat{\mu}(S)$ of anti-caches $S'$ such that $|S' \cap P_{\geq i}| < Y_i$ (and thus $|S' \cap P_{\geq i}| \leq \lceil Y_i \rceil - 1$, as required).

**Existence of page to move.** After matching the aforementioned $S$ to some $S'$, we want to identify some page $p \in P_i$ such that $p \in S \setminus S'$, so we can move it from the measure of $S$ to the measure of $S'$. Indeed, from the choice of $S$ and $S'$, it holds that $|S \cap P_{\geq i}| \geq \lceil Y_i \rceil + 1 \geq |S' \cap P_{\geq i}| + 2$. But, from the induction hypothesis for Iteration $i$, class $i + 1$ was balanced in $\mu_i$, and thus remains balanced at every step during Iteration $i$ (as this iteration never moves pages of classes $i + 1$ and above). This implies that $|S \cap P_{\geq i+1}| \leq \lceil Y_{i+1} \rceil \leq |S' \cap P_{\geq i+1}| + 1$. We can thus conclude that there exists $p \in P_i \cap (S \setminus S')$ as required.

**Balanced property.** Next, we prove that in $\mu_{i-1}$ after Iteration $i$, class $i$ is balanced, and the imbalance of any class $j < i$ is at most $\epsilon \cdot 3^{j_{\max}-(i-1)}$. Consider any step in Iteration $i$, where a measure $x$ of a violating state is identified; then, a page is moved from a measure $x$ to another measure $x$. The induction hypothesis for Iteration $i$ implies that the imbalance of class $i$ at $\mu_i$ is at most $\epsilon \cdot 3^{j_{\max}-i}$. Note that:

1. This step decreases the imbalance of class $i$ by at least $x$, as it decreases the imbalance in the violating state, but does not increase imbalance in the matched measure.

2. This step can increase the imbalance of a class $j < i$ by at most $2x$, in the worst case in which moving the page increased imbalance in both measures of $x$.

As a result, we can conclude that for $\mu_{i-1}$, at the end of iteration $i$, class $i$ is balanced, while the imbalance of every class $j < i$ increased by at most $2 \cdot \epsilon \cdot 3^{j_{\max}-i}$. Combining this with the hypothesis for iteration $i$, the imbalance of every class $j$ at $\mu_{i-1}$ is at most $\epsilon \cdot 3 \cdot 3^{j_{\max}-i} = \epsilon \cdot 3^{j_{\max}-(i-1)}$, as required.

This concludes the inductive proof of the claim.

**Cost analysis.** As mentioned before, every step in Iteration $i$ reduces imbalance at class $i$ by (at least) $x$, where $x$ is the measure of the chosen violating anti-cache state. The cost of this step is the cost of evicting a single page in $P_i$ from a measure $x$; as we assume that the weight of a page is at most its UCB, this cost is at most $x \cdot 6^{i+1}$. The inductive claim above states that the imbalance of class $i$ at the beginning of Iteration $i$ is at most $\epsilon \cdot 3^{j_{\max}-i}$; thus, the total cost of Iteration $i$ is at most $\epsilon \cdot 3^{j_{\max}-i} \cdot 6^{i+1} = \epsilon \cdot 6^{j_{\max}+1}/2^{j_{\max}-i}$. Summing over iterations, the total cost of REBALANCESUBSETS

is at most:

$$\sum_{i=j_{\min}}^{j_{\max}} \epsilon \cdot 6^{j_{\max}+1}/2^{j_{\max}-i} \le 12\epsilon \cdot 6^{j_{\max}}.$$

□

We can now prove that the distribution is consistent and balanced.

**Lemma B.4.** *The distribution maintained by Algorithm 2 is both consistent and balanced.*

*Proof.* First, we prove that the distribution is consistent. Indeed, note that consistency is explicitly maintained in Lines 9 and 12, and that Lemma B.3 implies that the subsequent calls to REBALANCESUBSETS does not affect this consistency.

As the distribution is consistent at any point in time, Lemma B.3 also implies that it is balanced immediately after every all to REBALANCESUBSETS; as the handling of every request ends with such a call, the distribution is always balanced after every request. □

At this point, we've shown that Algorithm 2 is legal: it maintains a valid distribution (through Lemma B.4 and Remark 5.3), and it provides samples to ONF when required (Proposition B.1); thus, it is a valid randomized algorithm for OWP-UW. It remains to bound the expected cost of Algorithm 2, thus proving Lemma 3.2; recall that this bound is in terms of $\overline{\text{ONF}}$, the cost of the fractional algorithm in terms of its UCBs rather than actual page weights.

*Proof of Lemma 3.2.* We consider the eviction costs incurred by Algorithm 2, and bound their costs individually.

First, consider the eviction cost due to maintaining consistency Line 9 (note that Line 12 only fetches pages, and incurs no cost). An increase of $\epsilon$ in $y_{p'}$ causes an eviction of $p'$ with $\epsilon$ probability; the expected cost of $\epsilon \cdot w_{p'}$ can be charged to the eviction cost of $\epsilon \cdot \text{UCB}_{p'}$ incurred in $\overline{\text{ONF}}$, and thus the overall cost due to this line is at most $\overline{\text{ONF}}$.

Next, consider the cost due to eviction during sampling (Line 17). Observe a page $p \in P$ that is evicted in this way; the cost of this eviction is $w_p$. For the first and second samples of $p$, we note that $w_p \le 1$; summing over $p \in P$, the overall cost of those evictions is at most $2n$. For subsequent samples of $p$, note that for $i > 2$, the $i$'th sample of $p$ is taken when $m_p \in (i-2, i-1]$. Thus, we can charge this sample to the fractional eviction that increased $m_p$ from $i-3$ to $i-2$, which costs $\text{UCB}_p$. Thus, the overall cost of this sampling is at most $\overline{\text{ONF}} + 2n$.

It remains to bound the cost of the REBALANCESUBSETS procedure. First, consider the cost of REBALANCESUBSETS due to sampling (Lines 16 and 20). Consider the state prior to such a call; some page $p$ has just been sampled, possibly decreasing $\text{UCB}_p$ and decreasing the class of page $p$, which could break the balanced property. Specifically, let $i, i'$ be the old and new classes of $p$, where $i' < i$. Then, imbalance could be created only in classes $j \in \{i'+1, \cdots, i\}$. In such class $j$, both $Y_j$ could decrease, and $|S \cap P_{\ge j}|$ could decrease for any anti-cache state $S$. However, as only one page changed class, one can note that the total imbalance in any such class $j$ is at most 1. Thus, Lemma B.3 guarantees that the total cost of REBALANCESUBSETS is at most $12 \cdot 6^i \le 12 \cdot \text{UCB}_p$. Using the same argument as for the cost of sampling, the total cost of such calls is at most $12\overline{\text{ONF}} + 24n$.

Now, consider a call to REBALANCESUBSETS in Line 10; A page $p'$ was evicted for fraction $\epsilon$ in ONF, and an $\epsilon$ measure of $p'$ was evicted in the distribution. Let $j$ be the class of $p'$; there could only be imbalance in classes at most $j$. For any such $i \le j$, $Y_i$ increased by $\epsilon$, and $|S \cap P_{\ge i}|$ increased by 1 in at most $\epsilon$ measure of states $S$. In addition, let $Y_i^-, Y_i^+ := Y_i^-$ be the old and new values of $Y_i$. Consider the imbalance in class $i$:

1. If $\lfloor Y_i^+ \rfloor = \lfloor Y_i^- \rfloor + 1$, then the imbalance of class $i$ can increase due to states $S$ where $|S \cap P_{\ge i}| = \lfloor Y_i^- \rfloor$ becoming unbalanced. But, due to consistency, the fact that $Y_i^- \ge \lceil Y_i^- \rceil - \epsilon$ implies that the measure of such pages is at most $\epsilon$; thus, the imbalance grows by at most $\epsilon$.

2. The adding of page $p'$ to an $\epsilon$-measure of pages can add an imbalance of at most $\epsilon$.

Overall, the imbalance in classes at most $j$ prior to calling REBALANCESUBSETS is at most $2\epsilon$. Through Lemma B.3, the cost of REBALANCESUBSETS is thus at most $24\epsilon \cdot 6^j$, which is at most $24\epsilon \cdot \text{UCB}_{p'}$. But, the increase in $\overline{\text{ONF}}$ due to the fractional eviction is at least $\epsilon \cdot \text{UCB}_{p'}$; thus, the overall cost of REBALANCESUBSETS called in Line 10 is at most $24 \cdot \overline{\text{ONF}}$.

Finally, consider a call to REBALANCESUBSETS in Line 13, called upon a decrease in $y_{p'}$ of $\epsilon$ for some page $p'$. Using an identical argument to the case of eviction, we can bound the cost of this call by 24 times the "fetching cost" of $\epsilon \cdot \text{UCB}_{p'}$. Now, note that the fractional fetching of a page exceeds the fractional eviction by at most 1; thus, the total cost of such calls is at most $24\overline{\text{ONF}} + 24n$.

Summing all costs, the total expected cost of the algorithm is at most $62\overline{\text{ONF}} + 50n$. □

## B.2 Proof of Theorem 1.1

We now combine the ingredients in this paper to prove the main competitiveness bound for Algorithm 2.

*Proof of Theorem 1.1.* First, assume the good event $\mathcal{E}$. Combining Lemma 3.2, Lemma 3.1 and Lemma C.2, it holds that

$$\mathbb{E}[\text{ON}] \leq O(\log k) \cdot \text{OPT}(Q) + \tilde{O}(\sqrt{nT})$$

Next, assume that $\mathcal{E}$ does not occur. Upper bound the cost of the algorithm by $O(n)$ per request, as in the worst case, the algorithm replaces the entire cache and samples each page in $P$ once. Thus, an upper bound for the cost of the algorithm is $O(nT)$. But, through Lemma C.1, the probability of $\mathcal{E}$ not occurring is at most $\frac{1}{nT}$. Thus, we can bound the expected cost of the algorithm as follows.

$$\mathbb{E}[\text{ON}] \leq \Pr[\mathcal{E}] \cdot \left( O(\log k) \cdot \text{OPT}(Q) + \tilde{O}(\sqrt{nT}) \right) + \Pr[\neg\mathcal{E}] \cdot O(nT)$$

$$\leq O(\log k) \cdot \text{OPT}(Q) + \tilde{O}(\sqrt{nT}).$$

□

## C Choosing Confidence Bounds and Bounding Regret

In this section we define the UCBs and LCBs, prove that they hold with high probability and bound the regret term.

Let $w_p^i$ be the $i$-th sample of page $p$. For the initial confidence bounds, we sample each page once and set $\text{LCB}_{p,1} = \frac{1}{2n^2T} \cdot w_p^1$, $\text{UCB}_{p,1} = 1$. Once we have $i > 1$ samples of page $p$, we define the confidence bounds as follows. Let $\bar{w}_{p,i} := \frac{1}{i} \sum_{j=1}^{i} w_p^j$ be the average observed weight, and $\epsilon_{p,i} = \sqrt{\frac{\log(4n^3T^3)}{2i}}$ be the confidence radius. Then, we set $\text{LCB}_{p,i} = \max\{\text{LCB}_{p,i-1}, \bar{w}_{p,i} - \epsilon_{p,i}\}$ and $\text{UCB}_{p,i} = \min\{\text{UCB}_{p,i-1}, \bar{w}_{p,i} + \epsilon_{p,i}\}$. The following procedure updates the confidence bounds online.

We show that the confidence bounds indeed bound the true weights with high probability (Lemma C.1), and then bound the regret term (Lemma C.2).

**Lemma C.1.** *Let $n_p$ be the final number of samples collected for page $p$. With probability at least $1 - \frac{1}{nT}$, the following properties hold.*

1. $0 < \text{LCB}_{p,1} \leq \text{LCB}_{p,2} \leq \ldots \leq \text{LCB}_{p,n_p} \leq w_p$ *for every page $p$.*

2. $w_p \leq \text{UCB}_{p,n_p} \leq \text{UCB}_{p,n_p-1} \leq \ldots \leq \text{UCB}_{p,1} = 1$ *for every page $p$.*

3. $\text{UCB}_{p,i} - \text{LCB}_{p,i} \leq 2\epsilon_{p,i}$ *for every page $p$ and $i \in [1, n_p]$.*

*Proof.* By definition, the LCBs are monotonically non-decreasing and the UCBs are monotonically non-increasing. Moreover,

$$\text{UCB}_{p,i} - \text{LCB}_{p,i} = \min\{\text{UCB}_{p,i-1}, \bar{w}_{p,i} + \epsilon_{p,i}\} - \max\{\text{LCB}_{p,i-1}, \bar{w}_{p,i} - \epsilon_{p,i}\}$$

$$\leq (\bar{w}_{p,i} + \epsilon_{p,i}) - (\bar{w}_{p,i} - \epsilon_{p,i}) = 2\epsilon_{p,i}.$$

---

**Algorithm 4:** Optimistic-Pessimistic Weights Estimation Procedure

---

**1** **Function** UPDATECONFBOUNDS$(p, \widetilde{w}_p)$ // update confidence bounds for $p$ upon new sample $\tilde{w}_p$.

**2**   **if** *this is the first sample of $p$ (i.e., $n_p = 1$)* **then**

**3**       define $\bar{w}_p \leftarrow \widetilde{w}_p$.

**4**       define $\mathrm{LCB}_p \leftarrow \frac{1}{2n^2T} \cdot \bar{w}_p$.

**5**       define $\mathrm{UCB}_p \leftarrow 1$.

**6**   **else**

**7**       update mean estimation $\bar{w}_p \leftarrow \frac{(n_p-1)\bar{w}_p + \widetilde{w}_p}{n_p}$.

**8**       define $\epsilon_p \leftarrow \sqrt{\frac{\log(4n^3T^3)}{2n_p}}$.

**9**       set $\mathrm{LCB}_p \leftarrow \max\{\mathrm{LCB}_p, \bar{w}_p - \epsilon_p\}$.

**10**       set $\mathrm{UCB}_p \leftarrow \min\{\mathrm{UCB}_p, \bar{w}_p + \epsilon_p\}$.

---

Thus, it remains to show that $\frac{1}{2n^2T} \cdot w_p^1 \le w_p$ and that $|w_p - \bar{w}_{p,i}| \le \epsilon_{p,i}$ for every page $p$ and $i \in [1, T]$ with probability $1 - \frac{1}{nT}$. For the first event, by Markov inequality and a union bound over all the pages $p \in P$, we have

$$\mathbb{P}\left[\exists p \in P. \ \frac{1}{2n^2T} \cdot w_p^1 > w_p\right] = \mathbb{P}[\exists p \in P. \ w_p^1 > 2n^2T \cdot w_p] \le n \cdot \frac{1}{2n^2T} = \frac{1}{2nT}.$$

For the second event, by Hoeffding inequality and a union bound over all the pages $p \in P$, all the time steps $t \in [1, T]$ and all the possible number of samples for each page $i \in [1, nT]$, we have

$$\mathbb{P}\left[\exists p \in P, i \in [1, T]. \ |w_p - \bar{w}_{p,i}| > \epsilon_{p,i}\right] \le n^2T^2 \cdot 2e^{-2i\epsilon_{p,i}^2} = n^2T^2 \cdot \frac{1}{2n^3T^3} = \frac{1}{2nT}.$$

The proof is now finished by taking a union bound over these two events. $\qquad\square$

Define $\mathcal{U} := \sum_{p \in P} \sum_{i=1}^{n_p} (\mathrm{UCB}_{p,i} - \mathrm{LCB}_{p,i}) + 2\log(1 + 1/\eta) \sum_{p \in P} \mathrm{LCB}_p$, the regret term used in Lemma 4.1.

**Lemma C.2.** *Under the good event of Lemma C.1, it holds that*

$$\mathcal{U} \le 8\sqrt{nT}\log(nT) = \tilde{O}(\sqrt{nT}).$$

*Proof.* The following holds under the good event.

$$\mathcal{U} = \sum_{p \in P} \sum_{i=1}^{n_p} (\mathrm{UCB}_{p,i} - \mathrm{LCB}_{p,i}) + 2\log(1 + 1/\eta) \sum_{p \in P} \mathrm{LCB}_p$$

$$\le 2 \sum_{p \in P} \sum_{i=1}^{n_p} \epsilon_{p,i} + 2\log(1 + 1/\eta) \sum_{p \in P} w_p \qquad\qquad \text{(Lemma C.1)}$$

$$= 2 \sum_{p \in P} \sum_{i=1}^{n_p} \sqrt{\frac{\log(4n^3T^3)}{2i}} + 2\log(1 + 1/\eta) \sum_{p \in P} w_p$$

$$\le \sqrt{2\log(4n^3T^3)} \sum_{p \in P} \sum_{i=1}^{n_p} \frac{1}{\sqrt{i}} + 2n\log(1 + 1/\eta) \qquad\qquad (w_p \le 1)$$

$$\le 2\sqrt{2\log(4n^3T^3)} \sum_{p \in P} \sqrt{n_p} + 2n\log(1 + 1/\eta) \qquad\qquad (\sum_{i=1}^{t} \frac{1}{\sqrt{i}} \le 2\sqrt{t})$$

$$\le 2\sqrt{2nT\log(4n^3T^3)} + 2n\log(1 + k),$$

where the last inequality holds by Cauchy–Schwarz and our choice of $\eta = 1/k$. Lastly, the stated upper bound follows since $k \le n$ and $n \le \sqrt{nT}$. $\qquad\square$

