# OpenReview forum: "Online Weighted Paging with Unknown Weights"
_NeurIPS.cc/2024/Conference — NeurIPS 2024 poster_

### Official Review · Reviewer_VdxB · 2024-07-10

**Soundness:** 3
**Presentation:** 3
**Contribution:** 3
**Rating:** 7
**Confidence:** 4

**Summary:**

This paper considers a generalization of the classical online paging problem. In classical online paging, one needs to maintain a cache of k slots as requests for fetching pages arrive online. If the requested pages are in the cache, there is no charged cost; otherwise, w_p cost will be charged for page p. The goal is to maintain a cache of k slots such that the total fetching cost is minimized. This paper extends this classical setting to the case where the weight of all pages is not known prior to algorithms. Instead, the algorithm needs to sample a value from an unknown distribution, and such a value will be an estimated value of the actual weight. The goal is still to minimize the total fetching cost.

The main contribution of this work is an online algorithm that achieves alg$\leq O(\log k)$ opt + $O(\sqrt{nT})$, where k is the size of the cache, n is the number of pages and T is the number requests. The main technique is fractional solution+rounding. To bridge these two phases, the authors design an interface, which aims to learn an estimation of weights for each page.

**Strengths:**

1. The studied problem is well-motivated and it should be interested in the ML community. I expect that the studied problem will have a positive impact on practice.

2. I like the interface idea to bridge the fractional solution and rounding. The main purpose of the interface is to learn the weights. Although all these ideas are not novel, it is interesting to see that a simple combination works well.

3. The paper is well-structured. I appreciate that the authors give a very clear statement for the algorithmic framework in Section 3. This is very helpful for readers.

**Weaknesses:**

There is no lower bound in the paper. The regret bound might be far from tight, but this is expected in the first paper.

**Questions:**

Could you comment on how tight the regret bound O(\sqrt{nT}) is? This seems to be a large number.

**Limitations:**

This is a theoretical paper, there is no potential negative societal impact.

---

> ### Author Rebuttal · Authors · 2024-08-05
>
> We would like to thank the reviewer for the thoughtful comments and encouraging review.
>
> Regarding lower bounds, see the remark in the general rebuttal. In essence, we’ve presented lower bounds for the problem in “Our Results”, which we intend to further formalize in the camera-ready version.
>
> Regarding the regret bound of $O(\sqrt{nT})$, consider the following implications of the lower bounds (a) and (b) that appear in this paragraph (starting at Line 61 in the paper):
> 1. Lower bound (b) implies that allowing a competitive ratio that is $o(\log k)$, the regret of any algorithm is $\Omega(T/k)$. In particular, for constant k, this implies $\Omega(T)$ regret. Without loss of generality $n\le T$, as pages that are never requested are never loaded into the cache; thus, the regret lower bound is $\Omega(\sqrt{nT})$ in this case.
> 2. The term $\Omega(\sqrt{T})$ is necessary for any choice of (sublinear) competitive ratio.
>
>
> We thank the reviewer again for the supportive review and will happily address any further questions.

---

> > ### Comment · Reviewer_VdxB · 2024-08-14
> >
> > I'd like to thank the authors for addressing my concerns. The rebuttal addresses my concerns and I will keep my score.

---

### Official Review · Reviewer_MmEo · 2024-07-11

**Soundness:** 3
**Presentation:** 2
**Contribution:** 3
**Rating:** 6
**Confidence:** 4

**Summary:**

This paper is the first to model and study to the online weighted paging problem with unknown weights. In this model, the weights $w_p$ of pages are initially unknown, and the eviction cost is drawn from an unknown distribution with an expectation equal to $w_p$. This study extends previous research on online weighted paging with unknown weights by integrating concepts from online learning.

The authors present an algorithm that achieves the performance bound as follows:
$$
E[ALG]\ \leq O(\log k) OPT + \tilde{O}(\sqrt{nT}).
$$
Remark that $O(\log k) is the competitive ratio in the known-weight setting. The proposed solution comprises two main components: a fractional algorithm and a rounding framework. The fractional algorithm is designed to approximate the optimal cost, while the rounding framework ensures consistency with the fractional solution through a rebalancing routine. The methodology for the fractional algorithm and the rounding and rebalancing framework are from [Bansal et al. 2010] and [Bansal et al. 2012]. Additionally, the weight estimation utilizes a classical Upper Confidence Bound (UCB) method from the multi-armed bandit problem in online learning. It is a non-trivial combination of the two techniques.

**Strengths:**

This paper presents their motivation for studying the new model clearly, using the example of multi-level cache structure as a compelling rationale. The proposed model is clean, providing a robust foundation that is likely to inspire future research.

**Weaknesses:**

The presentation in section 5 is not clear and may only be reader-friendly to those familiar with it [Bansal et al. 2012]. We may need a high-level idea and some introductions, like anti-cache, before presenting the pseudocode.

**Questions:**

In section 5, the author mentions that they need a more robust rebalancing procedure. What is the detailed difference between it and the previous approach? Can you give some more explanations?

It seems that the algorithm runs really slow because of the continuous rebalancing step. Can you remark on the running time of the algorithm?

**Limitations:**

The author does not address the limitations of the proposed algorithm. The efficiency of the algorithm is not considered. Given that the algorithm maintains an exponential number of subsets $S \subset P$ and may adjust some $\mu(S)$ each time an $\epsilon$ fraction of $\mu(S)$ changes, the running time may be quite large.

---

> ### Author Rebuttal · Authors · 2024-08-05
>
> We thank the reviewer for the thoughtful comments and supportive review.
>
> We take into account the comment regarding the presentation in Section 5. In the final version of the paper, we will edit this section to be more clear and emphasize high-level ideas.
>
> Regarding your question about our rebalancing procedure, this procedure is different from that of Bansal, Buchbinder and Naor [2012] in the following ways:
> 1. Their procedure had to maintain the balanced property only upon changes to the distribution towards maintaining consistency (e.g., a certain probability mass gaining/losing a page). Our rebalancing procedure must also handle the case in which pages change class due to changes in their UCB (as page classes are not constant in our algorithm).
> 2. A technical observation: note that the rebalancing procedure fixes the imbalance at every level, in descending order. (The imbalance is the striped area in Figure 2.) However, fixing the imbalance at level $i$ can increase the imbalance at levels $\le i-1$. In BBN, after fixing an imbalance of $x$ at level $i$, the imbalance at levels $\le i-1$ can be seen to be at most $x$. However, in our case, the imbalance at levels $\le i-1$ can be at most $3x$; i.e., the imbalance grows exponentially. This tougher case of cascading imbalance occurs when fixing imbalance due to a page-changing UCB class.
>
> Regarding the mentioned limitation, the rounding scheme does maintain a distribution supported by many cache states (while, of course, holding in actuality a single cache state). To the best of our knowledge, this is true of any known rounding scheme for weighted caching, in particular for that of Bansal, Buchbinder, and Naor [2012]. The focus of this paper was introducing unknown weights to weighted paging, rather than improving the computational tractability of the rounding scheme for known weights; however, we agree with your observation and believe this is a promising direction for future study.
>
> We thank you again for the thoughtful review. We would be happy to discuss any further notes/questions you have. If we’ve properly addressed your concerns, we would be happy if you considered updating your assessment accordingly.

---

### Official Review · Reviewer_WtQQ · 2024-07-11

**Soundness:** 4
**Presentation:** 3
**Contribution:** 4
**Rating:** 7
**Confidence:** 4

**Summary:**

The paper addresses the problem of online weighted paging with eviction costs drawn from unknown page-dependent distributions. As evictions occur, the authors demonstrate how to learn an effective eviction strategy online using previous cost samples, framing this as a multi-armed bandit problem. They first approach a fractional relaxation of the problem, then apply a rounding technique to derive a randomized algorithm for the original problem. The resulting algorithm incurs an expected cost of $O(\log k) \cdot \text{OPT} + O(\sqrt{nT})$, where $k$ is the cache size, $n$ is the number of pages, and $T$ is the number of time steps.

**Strengths:**

* The paper is well-written and well-organized.
* The setting is both well-motivated and theoretically interesting, as it combines techniques from competitive analysis and regret analysis.
* The main result is strong and might inspire future work in the field of online algorithm design.

**Weaknesses:**

I do not see any major weaknesses in this paper, but there are a few minor ones:
* Although the setting is different, the paper is closely related to the framework of learning-augmented algorithms. I suggest that the authors mention this connection in the related work section.
* A very similar setting has been studied in "On Preemption and Learning in Stochastic Scheduling" (ICML 2023) for the non-clairvoyant scheduling problem, which also couples the analyses of the competitive ratio and regret. Maybe this should be cited as a related work.
* While the main contribution is theoretical, it would be nice to conduct a few experiments to demonstrate how the proposed algorithms compare to other benchmark algorithms where the weights are known

**Questions:**

Since the algorithm only chooses which of the $k$ pages in the cache to evict, would it be possible to improve the regret term in Theorem 1.1 and have instead a term of the form $O(\sqrt{f(k,n) T})$, with $k \leq f(k,n) < n$? If not, can you prove that the bound $O(\sqrt{nT})$ is the best possible for any values of $n$ and $k$?

**Limitations:**

The assumptions of the theorems are clearly stated.

---

> ### Author Rebuttal · Authors · 2024-08-05
>
> We would like to thank the reviewer for the encouraging review, as well as the thoughtful comments. Please see below our response regarding the weaknesses and questions mentioned.
>
> _“… the paper is closely related to the framework of learning-augmented algorithms. I suggest that the authors mention this connection in the related work section.”_ \
> Thanks for this remark, we’ll explore the connection between this problem and learning-augmented algorithms in the related work section.
>
> _“...it would be nice to conduct a few experiments…”_ \
> We’ll consider adding an experimental section towards demonstrating the applicability of the algorithm. We note that in bandit papers that focus on proving regret bounds, experimental sections are usually optional. (Unlike, e.g., learning-augmented online algorithms, where empirical evaluations are customary for ML conferences.)
>
> _“Since the algorithm only chooses which of the 𝑘 pages in the cache to evict, would it be possible to improve the regret term in Theorem 1.1 and have instead a term of the form $𝑂(\sqrt{𝑓(𝑘,𝑛)𝑇})$, with $𝑘≤𝑓(𝑘,𝑛)<𝑛$? If not, can you prove that the bound $𝑂(\sqrt{𝑛𝑇})$ is the best possible for any values of 𝑛 and 𝑘?”_ \
> As mentioned in the lower bounds described in “our results”, in the regime that allows a competitive ratio of $\Theta(log k))$, there is a lower bound on regret of only  $\Omega(\sqrt{T})$. It could be the case that the $\sqrt{n}$ term in our $O(\sqrt{nT})$ regret bound could be improved upon; as this is the first work on this problem, we did not focus on obtaining tight bounds. However, we remark that $\Theta(\sqrt{nT})$ is the optimal regret bound for standard MAB, and we would thus be surprised if a significant gap were obtained for OWP-UW.
>
> We thank you again for your positive assessment and hope the above has satisfied your concerns. We would be happy to address any further questions you may have.

---

> > ### Comment · Reviewer_WtQQ · 2024-08-07
> >
> > I thank the authors for their response, and I strongly encourage them to add the necessary discussion on the relation with learning-augmented algorithms, and in particular the works where the unknown variables are learned during the algorithm execution.
> >
> > I would have liked to see experiments mostly because I am curious to observe if the asymptotic behavior, when T is very large, aligns with that of prior algorithms that have knowledge of the weights, in terms of the multiplicative constant in $O(\log k)$. However, I agree that this is not a major weakness of the paper, but a suggestion instead, as the theoretical contribution is sufficiently interesting.

---

### Official Review · Reviewer_qj5j · 2024-07-12

**Soundness:** 2
**Presentation:** 2
**Contribution:** 3
**Rating:** 4
**Confidence:** 2

**Summary:**

The paper studies a the online paging problem where the weights to retrieve a items are independent random variables with potentially different distributions. The paper introduces an algorithm whose expected performance differ from the optimal one by a multiplicative factor (logarithmic in the size of the cache) and an additive term (sublinear in the number of the requests).

**Strengths:**

If correct, the paper makes a very interesting advance on an important theoretical topic. The authors position well the paper in the existing literature.

**Weaknesses:**

I found this paper the most difficult to evaluate. I feel NeurIPS may not be the most suited venue for such work, both for its theoretical focus (it seems more a STOC-paper) and the 9-page limit which forced the authors to compress much their reasoning. Probably for this reason, as well as some disorganization in the presentation, I was not able to figure out how the algorithm really works as well as to follow some more technical arguments. I provide some specific comments below

Doubts
- about algorithm 1, what is really behind "continually increase" at line 4, should we assume that that y_p and m_p are updated according to differential equations? This seems to be supported also by later statements like at line 276 "upon any (infinitesimally small) change to a fractional variable"
- I was not able to understand the reasoning about why the problem may not exhibit sublinear regret without a competitive ratio (point b at page 2)
- in section 1.2, the paper argues that the solution cannot be built in the standard way by combining a solution to the fractional problem and a rounding scheme. I was not able to follow the reasoning in lines 86-94. Even more, because the proposed solution combines a solution to the fractional problem and a rounding scheme
- in footnote 2, "note that k \le \sqrt{nT}", I did not get if this follows from the previous reasoning or it is an implicit assumption that T is large enough.
- algorithm 2, what does it mean line 9 "add p' to the anti-cache in an \epsilon-measure of states without p'"? If find this kind of expressions too vague.
- lines 254-255, there is probably an error, how can be UCB_p between 6^i and 6^{i+1} if, as observed in the same line UCB_p is between 0 and 1?

Motivation
- I find the motivation in terms of a cache inserted in the a hierarchy of caches quite weak. The content stored at caches of higher level would be determined by the caching decisions at lower levels. It would the not lead to the random weights considered in this paper.

Confused presentation
- the fact that costs may be considered to be paid upon eviction rather than upon fetching is mentioned in footnote 2 at page 4, but it is already implicitly used at page 2 when explaining why the problem cannot admit logarithmic competitive ratio without regret
- the paper starts talking about moving fraction and servers at line 166 before referring to the parallel with the k-server problem
- the (huge) caption of figure 1 refers to the "consistency property" and the "subset property". The first one is never really formally defined. The second one is only introduced two page later.
- in the caption of figure 2, there is a reference to "class i" but this is only introduced at page 8. Also note that there are two classes i P_i and P_{\ge i}



Minor
- the problem is sometimes referred to as OWP-UW, others as UW-OWP
- line 95: "the and rounding scheme"
- footnote 1: "known in advanced"
- in the figures the lables of the different subfigures (e.g. a, b, c, d in figure 1 are missing)

Update after rebuttal
- the authors have addressed some of my technical questions and I have increased the score.

**Questions:**

See questions above.

**Limitations:**

The computational complexity of the algorithm is not discussed. This is important, in particular if, as I think, the algorithm relies on solving some differential equations.

---

> ### Author Rebuttal · Authors · 2024-08-05
>
> We would like to thank the reviewer for their thorough assessment and comments. Here is our response to the points raised.
>
> **Suitability for NeurIPS**: we note that theory papers considering online caching/paging have appeared in ML conferences when combined with an ML-related theme. See for example [1,2,3,4,5], which are theory papers that augment caching/paging with ML predictions, and have appeared in ICML. In our case, multi-armed bandits are clearly appropriate for NeurIPS (in fact, they constitute a primary area). Thus, we believe that NeurIPS is the appropriate venue for this paper.
>
> **Computational tractability**: please see the general rebuttal. In short, the continuous presentation of the algorithm (also chosen for previous work) can easily be replaced with a computationally efficient discrete process.
>
> **Regret lower bound**: the described construction implies that for every randomized algorithm $ALG$ there exists an input such that $OPT = O(T/(k \log k))$ and $E[ALG]=\Omega(T/k)$. Thus, $E[ALG]-OPT$ (i.e., the regret) is at least $\Omega(T/k)$; that is, the regret grows linearly in $T$.  We intend to formalize this further in the camera-ready version; see general rebuttal.
>
> **Fractional+rounding framework**: The point in Lines 86-94 was that in most algorithms involving fractional+rounding schemes, the integral problem immediately induces a fractional problem, such that the fractional solver is competitive w.r.t. this fractional problem, independently of any “downstream” rounding scheme. This is the case for Bansal, Buchbinder and Naor’s fractional solver for online paging with known weights. But, for unknown weights, no fractional solver can be competitive without learning about page weights – and this is done through sampling pages integrally, which cannot be done by a fractional solver. Thus, the interaction between the fractional solver and the rounding scheme is crucial even for the fractional solver itself to have any competitiveness guarantee. Indeed, one of the main contributions of this paper is devising an interface between the fractional solver and the rounding scheme that enables sampling page weights when needed.
> Thank you for this input regarding writing clarity, we’ll improve this paragraph in the camera-ready version.
>
> **Additional notes**:
> 1. _" note that $k \le \sqrt{nT}$"_: as noted in Line 146 in the preliminaries section, it holds that $k<n$ (otherwise the problem is trivial as all pages can be simultaneously held in the cache). Also, as $n$ is the number of requested pages and thus $n \le T$, it also holds that $k<T$. The observation follows.
> 2. _"add $p'$ to the anti-cache in an \epsilon-measure of states without $p'$"_: In keeping with previous work on online weighted paging, our randomized algorithm is described as maintaining a distribution over cache states at any point in time; of course, in practice, only a single state is held by the algorithm and is updated as the distribution changes. For example, if the algorithm holds anti-cache state $S$ which has measure $x$ in the distribution, and the distribution is then updated such that $y<x$ measure of state $S$ receives page $p’$, then page $p’$ will be added to the anti-cache state with probability $y/x$ to maintain consistency.
> While this is consistent with previous work, we agree that this should be made more explicit; we will do so in the camera-ready version.
> 3. Lines 254-255: there is no mistake here. The classes of values between $0$ and $1$ correspond to non-positive values of $i$.
>
> As for the other remarks regarding writing, thanks for bringing those to our attention; we will modify the paper accordingly for the final version.
> We hope the above has satisfied your concerns and questions, and that you positively consider increasing your assessment.
> We, of course, will happily address any further questions you may have.
>
>
>
>
> [1] "Paging with Succinct Predictions", Antoniadis, Boyar, Eliáš, Favrholdt, Hoeksma, Larsen, Polak and Simon, ICML 2023
>
> [2] "Parsimonious Learning-Augmented Caching", Im, Kumar, Petety and Purohit, ICML 2022
>
> [3] "Robust Learning-Augmented Caching: An Experimental Study", Chłędowski, Polak, Szabucki and Żołna, ICML 2021
>
> [4] "Online metric algorithms with untrusted predictions", Antoniadis, Coester, Elias, Polak and Simon, ICML 2020
>
> [5] "Competitive Caching with Machine Learned Advice", Lykouris and Vassilvitskii, ICML 2018

---

> > ### Comment · Reviewer_qj5j · 2024-08-13
> >
> > I thank the authors for their explanations. I was convinced by the technical answers. I still maintain that the motivation is quite weak (and I went through the discussion with the reviewer Yutu), the main text is packing too much information, and the presentation is quite disorganized. I am (slightly) increasing the original score.

---

> > > ### Author Response · Authors · 2024-08-13
> > >
> > > We appreciate your feedback and would like to thank you for your consideration of our comments.
> > >
> > > Regarding motivation, as stated in our response to Reviewer Yutu, there are multiple motivations for studying OWP-UW, as in essence the deterministic-weights assumption in previous papers is often unjustified. One specific example is stated in our comment to Reviewer Yutu (handling stochastic weights, e.g., when retrieving data from the internet). In the final version of the paper, we'll be sure to expand our motivation segment; e.g.,  through including this additional motivating case.
> > >
> > > Regarding writing, we thank you again for your feedback. We'll ensure that our final version has a clear presentation that addresses your concerns.

---

### Official Review · Reviewer_Yutu · 2024-07-12

**Soundness:** 3
**Presentation:** 2
**Contribution:** 3
**Rating:** 4
**Confidence:** 4

**Summary:**

Author consider caching with stochastic weights. In particular,
the cost incurred by the algorithm at eviction of a page p
is drawn independently from a fixed distribution D_p
which is different for every page and is not known to the algorithm in advance.
Authors propose an algorithm with guarantees which combine both
competitive ratio and regret. Authors claim that neither regret nor competitive
ratio are possible on their own.
Their algorithm is a modification of the classical algorithm
by Bansal et al. for weighted paging, incorporating upper and lower confidence
bounds on the average weight of the page.
The more challenging part seems to be randomized rounding, where they
have to deal with the changes of the upper confidence bounds and such
element was not present in the original paper of Bansal et al.

**Strengths:**

* Problem seems interesting
* Result is non-trivial from technical point of view

**Weaknesses:**

* I find the framing of their result in the context of hierarchical caching
and CPU caches very unfortunate for several reasons:

(1) hierarchical caching is already studied (see work of Bansal, Buchbinder, Naor SICOMP'12). Authors do not seem to be aware of this.

(2) It is completely unclear why should there be an underlying distribution
of the cost of loading each page in a hierarchical (or multi-level) cache:
It is enough that the algorithm operating on different levels of the cache
are mis-aligned in some way to make weights of the pages
adversarial instead of stochastic.

(3) I cannot imagine implementing anything resembling their approach
in CPU caches.

* Provided justification of the optimality of the presented results
is very informal. It is just a few lines.
In particular, authors argue about the necessity of the regret term using using
lower bounds for bandits. However, their input is stochastic which
allows much better regret bounds when the mean weights of the pages
differ significantly. On the other hand, if the weights of the pages are
similar, it is easy to achieve good competitive ratio.

* Statements of their results are not completely clear: E.g., what is Q in
Theorem 1.1? Theorem says it is an "input":
Is it composed of the request sequence and the distribution of
page weights, or does it contain also the realization of the page weights?
What is OPT(Q) then? I do not see these things explained even in preliminaries.

**Questions:**

* Please comment on the last two weaknesses mentioned above.

**Limitations:**

Authors do not discuss possible difficulties in implementing their algorithm
in CPU caches although they use such use case as a main motivation.

---

> ### Author Rebuttal · Authors · 2024-08-05
>
> We thank the reviewer for the thoughtful feedback, please see below our comments for the concerns raised.
>
> **Regarding (1):** _“Hierarchical caching is already studied”_.
> Assuming you refer to [1]: this paper studies generalized weighted paging, in which each page has both a weight and a size demand imposed on the cache. Their only reference to storage hierarchies refers to weights (not sizes); thus, we interpret the reviewer’s comment as claiming that our motivating example is captured by standard weighted paging with known weights.
> To illustrate why this is not the case, note that in standard weighted paging, different pages have different weights, but each specific page has the same weight over time. In our motivating example, this is not the case: the same page can alternate between the main memory (fetching cost 1) and the L2 cache (fetching cost epsilon << 1). This motivates learning a page’s expected fetching cost over time, which is the crux of our model.
>
>
> **Regarding (2):**  _“...why should there be an underlying distribution of the cost of loading each page in a hierarchical cache”_.
> Our goal in this work is to remove the commonly used known-eviction-costs assumption in online weighted paging. Our motivating example refers to the aforementioned cache hierarchy, in which the L2 cache holds pages with probability proportional to their demand by the individual cores; we believe this is a reasonable assumption to make. We agree that the requested computation could in theory “game” the L2 cache; but, we don’t think encapsulating this (arguably somewhat pathological) case would make our motivating example more compelling.
>
> **Regarding (3):**
> (a) _“I cannot imagine implementing…”_.
>  This paper, as previous works in this research field, is theoretical. Our algorithm is not claimed to be practically implementable, which also resembles previous works.
> The only inefficient component in our algorithm is the rounding scheme, as it performs infinitesimal iterations, similar to the original rounding scheme proposed by Bansal, Buchbinder and Naor (FOCS’ 07).
> “Justification of the optimality … is very informal”. Our upper bound combines both the element of learning the page weight, which yields a regret term and the element of planning a close-to-optimal eviction strategy, which yields a competitive ratio term. We claimed that each term separately is optimal, and the combination of both is necessary. As you mentioned, our bounds provide a trade-off between the regret caused by the unknown costs and a competitive ratio w.r.t the optimal algorithm that knows the true expected eviction costs.
>
> (b) _“What is $Q$ in Theorem 1.1?”_.
> The overwhelming majority of work in randomized online algorithms deals with oblivious adversaries; this is also the case in this work. In particular, the input $Q$ consists of a sequence of $T$ requests for pages, where every page has an associated weight distribution; the oblivious adversary commits to this input. When the algorithm processes input $Q$, upon evicting a page, the algorithm obtains an independent sample from the weight distribution of that page.
>
> We would be happy to elaborate on any issue during the discussion phase. If we’ve addressed your remarks to your satisfaction, we would be happy if you considered revising your merit score accordingly.
>
> [1] "Randomized competitive algorithms for generalized caching", Bansal, Buchbinder & Naor, 2012.

---

> > ### Comment · Reviewer_Yutu · 2024-08-09
> >
> > * reference [1] shows how to model the hierarchical cache where the cost of a page load depends on what layer of the cache contains the page as a generalized caching problem.
> >
> > * what algorithm do you expect to be maintaining the L2 cache? Which algorithm satisfies the property that a given page is in the cache with some probability dependent only on the page itself? I do not find expecting such property reasonable at all. Does your algorithm satisfy this property?
> >
> > * Yes, there are many theoretical papers on caching but it is definitely not common motivating a complicated and resource demanding algorithm by CPU caches.
> >
> > to sum up, I am still not happy about the framing of your result.
> >
> > Now about the tightness of your result. You provide a bound which combines a competitive ratio with a regret term depending on T.  You claim that sometimes you may need the regret term and sometimes competitive ratio. But your bound contains both at the same time, which  is pretty weak in the context of the competitive analysis as well regret analysis. Do you have a lower bound showing that you need both terms at the time? I have explained my doubts about the existence of such bound in my review.
> >
> > thank you for explaining the statement of your theorem. your problem is partially adversarial and partially stochastic which is not a particularly common setting. I believe that this deserves a proper explanation in the paper.

---

> > > ### Author Response · Authors · 2024-08-09
> > >
> > > Thank you for your comment,
> > >
> > > Regarding motivation:
> > > 1. Suppose that the L2 cache fetches any page requested by a core, and evicts uniformly at random. Then, if a page is requested twice as often by the cores than another page, the probability of that page appearing in the cache at a given point in time is roughly twice as large.
> > > 2. Perhaps more importantly than the specifics of the CPU cache example, we note that the motivation for our work is not different from that of any paper in the significant line of work on weighted caching. In fact, the assumption made in these previous papers that the page weights are known is often unjustified. For example, as you've mentioned [1], they use as motivation hierarchical caching for web pages:
> > > *"This version of the problem is called weighted caching and it models scenarios in which the cost of fetching a page is not the same due to different locations of the pages (e.g., main memory, disk, Internet)"*. The cost of fetching a page from the internet cannot be assumed to be constant, and can be modeled much more realistically as drawn from some unknown distribution. This assumption made in all prior weighted paging papers ([1] included) is the one that our paper addresses.
> > >
> > > Regarding the lower bounds: \
> > > You have perhaps misunderstood the claim reflected in our lower bounds. We did *not* merely claim that either a competitive ratio or a regret term are needed. We claimed that (a) In the absence of a competitive ratio, a terrible regret term is needed (much worse than our regret bound), and (b) in the absence of a regret term, a terrible competitive ratio is needed (much worse than our competitive ratio). The conclusion to draw is that **both** a competitive ratio term and a regret term are needed. We will ensure that this point is emphasized well in the paper.
> > >
> > >
> > > *"Your problem is partially adversarial and partially stochastic which is not a particularly common setting"*: \
> > > In fact, the majority of works in the field of multiarmed bandits use this setting. For example, in arguably the most well-known variant of MAB, each arm has a value distribution chosen adversarially, while the samples from this distribution are obtained stochastically. We nevertheless value the writing feedback and will ensure that our paper makes this clear.

---

> > > > ### Comment · Reviewer_Yutu · 2024-08-12
> > > >
> > > > > Suppose that the L2 cache fetches any page requested by a core, and evicts uniformly at random. Then, if a page is requested twice as often by the cores than another page, the probability of that page appearing in the cache at a given point in time is roughly twice as large.
> > > >
> > > > this cannot be true unless you have some very strong assumptions about the request sequence. But the request sequence is adversarial in your setting, right?

---

> > > > > ### Author Response · Authors · 2024-08-12
> > > > >
> > > > > Our algorithm indeed handles even adversarial inputs, and does not rely on stochasticity in the request sequence. Of course, our algorithm can handle stochastic input sequences as well.
> > > > >
> > > > > Zooming out, it appears that your remaining concern is with the CPU cache motivating example. The goal of this example was merely demonstrating how sampled weighted instances can arise from seemingly unweighted instances. The motivation for studying OWP-UW is wider; in fact, we gave one such motivation in our previous comment regarding [1]. (We'll integrate this additional motivating case into the camera ready paper.)
> > > > >
> > > > > If indeed the only remaining concern is with the motivating paragraph, we hope that you'll consider revising your score accordingly. If there are any additional concerns, we'll be happy to address them.

---

### Author Rebuttal · Authors · 2024-08-05

We thank the reviewers for their thorough assessments and their thoughtful remarks. Here are some clarifications regarding themes that appear in more than one review.

**Running time:**

Reviewers Yutu and qj5j made remarks regarding the running time of the algorithm, and specifically the continuous updates in the fractional solver. The fractional solver is presented as continuously decreasing and increasing anti-server amounts, in keeping with previous work. But, this continuous process can be easily discretized into simple multiplicative updates, without affecting the correctness of the algorithm. This is a common trait of algorithms based on multiplicative updates; for an introduction to such discretization, see e.g. Chapter 4.2 of [1]. Similarly, the rounding scheme which maintains consistency and balance is presented as processing infinitesimal changes (in keeping with previous work), but can actually be applied only after the fractional solver has concluded; i.e., once per request.



**Lower bounds:**

Several reviewers had issues with the lower bounds complementing our main result. In particular, we showed that:
1. In the absence of a regret term, the competitive ratio of any algorithm is unfavorable (i.e., it has a polynomial dependence on $T$).
2. In the absence of a competitive ratio term, the regret of any algorithm is unfavorable (i.e., linear in $T$).
We concluded that both a competitive ratio term and a regret term are necessary. As these lower bounds are rather simple, we chose to present them informally in “our results”; however, these lower bounds are concrete, and can easily be formalized as theorems. Given the reviewers’ feedback, we’ll formalize those lower bounds in the camera-ready version.

[1] “The Design of Competitive Online Algorithms via a Primal-Dual Approach” by Niv Buchbinder and Seffi Naor.

---

### Comment · Area_Chair_245g · 2024-08-12
**Please comment on the rebuttal**

Dear reviewer qj5j, MmEo and VdxB,

Can you please read and comment on the rebuttal from the authors? Please also try to take other reviews and responses into consideration.

Regards,
Area Chair

---

### Decision · Program_Chairs · 2024-09-25

**Decision:**

Accept (poster)

**Comment:**

This paper suggested a stochastic weight setting for online weighted paging. The access to the weight is modeled as a multi-armed bandit  (MAB) model.

The review is somewhat mixed. it is generally agreed that the motivation of considering stochastic weight makes sense. However, the main criticism is that the specific model of MAB is not well justified, and the mere motivating example in the paper may not be sufficient. Nonetheless, it seems MAB is a simple yet powerful model, and it makes sense to start with this as one of the early paper to initiate the study of the stochastic setting.

The review somewhat agreed that the algorithm is technically nontrivial, and is an interesting combination of the online primal-dual/rounding framework with MAB. However, criticisms including practical efficiency of the algorithm and the clarity of the technical presentation (for example the vague lower bound argument) are also raised.

This is a borderline paper, but I suggest for an acceptance because of the conceptual value of initiating the stochastic setting of online weighted paging, as well as the technical strength of the paper.